

# Nitrous acid formation in a snow-free wintertime polluted rural area

Catalina Tsai[1], Max Spolaor[1], Santo Fedele Colosimo[1], Olga Pikelnaya[1,2], Ross Cheung[1], Eric Williams[3], Jessica B. Gilman[3], Brian M. Lerner[3,4], Robert J. Zamora[3], Carsten Warneke[3], James M. Roberts[3], Ravan Ahmadov[3], Joost de Gouw[5], Timothy Bates[6], Patricia K. Quinn[6], and Jochen Stutz[1]

[1]Department of Atmospheric and Oceanic Sciences, University of California Los Angeles, Los Angeles, CA, USA
[2]now at South-Coast Air Quality Management District, Diamond Bar, CA, USA
[3]Earth System Research Laboratory, NOAA, Boulder, CO, USA
[4]now at Aerodyne Research, Inc., Billerica, MA, USA
[5]Cooperative Institute for Research in Environmental Sciences, University of Colorado, Boulder, CO, USA
[6]Pacific Marine Environmental Laboratory, NOAA, Seattle, WA, USA

*Correspondence to:* Jochen Stutz (jochen@atmos.ucla.edu)

**Abstract.**

Nitrous acid (HONO) photolysis is an important source of hydroxyl radicals (OH) in the lower atmosphere, in particular in winter when other OH sources are less efficient. The nighttime formation of HONO and its photolysis in the early morning have long been recognized as an important contributor to the OH budget in polluted environments. Over the past few decades it has become clear that the formation of HONO during the day is an even larger contributor to the OH budget, and additionally provides a pathway to recycle $NO_x$. Despite the recognition of this unidentified HONO daytime source, the precise chemical mechanism remains elusive. A number of mechanisms have been proposed, including gas-phase, aerosol, and ground surface processes, to explain the elevated levels of daytime HONO. To identify the likely HONO formation mechanisms in a wintertime polluted rural environment we present LP-DOAS observations of HONO, $NO_2$, and $O_3$ on three absorption paths that cover altitude intervals from 2 m to 31 m, 45 m, and 68 m agl during the UBWOS 2012 experiment in the Uintah Basin, Utah, USA. Daytime HONO mixing ratios in the 2 - 31 m height interval were, on average, 78 ppt, which is lower than HONO levels measured in most polluted urban environments with similar $NO_2$ mixing ratios of 1-2 ppb. HONO surface fluxes at 16 m agl, calculated using the HONO gradients from the LP-DOAS and measured eddy diffusivity coefficient, show clear upward fluxes. The hourly average vertical HONO flux during sunny days followed solar irradiance, with a maximum of $(4.9\pm0.2)$ $\times 10^{10}$ molec. $cm^{-2}$ $s^{-1}$ at noontime. A photo-stationary state analysis of the HONO budget shows that the surface flux closes the HONO budget, accounting for $63\pm32\%$ of the unidentified HONO daytime source throughout the day and $90\pm30\%$ near noontime. This is also supported by 1D chemistry and transport model calculations that include the measured surface flux, thus clearly identifying chemistry at the ground as the missing daytime HONO source in this environment. Comparison between HONO surface flux, solar radiation, $NO_2$ and $HNO_3$ mixing ratios and results from 1D model runs suggest that, under high $NO_x$ conditions, HONO formation mechanisms related to solar radiation and $NO_2$ mixing ratios, such as photo-enhanced conversion of $NO_2$ on the ground, are most likely the source of daytime HONO. Under moderate to low $NO_2$ conditions, photolysis of $HNO_3$ on the ground seems to be the main source of HONO.



## 1 Introduction

HONO is an important precursor of OH in the atmosphere. Observations in urban and rural areas show that HONO photolysis can contribute up to 55% to the daytime radical budget, both from the photolysis of HONO accumulated throughout the night, as well as from the photolysis of HONO continuously formed throughout the day (e.g. Alicke et al., 2002, 2003; Kleffmann et al., 2005; Kleffmann, 2007; Ren et al., 2003; Volkamer et al., 2010; Young et al., 2012). The significance of HONO as an OH precursor varies depending on the environment, but it generally appears to be most prominent during winter, when other OH sources are less efficient (e.g. Ren et al., 2003; Elshorbany et al., 2009). HONO chemistry can also play a significant role in the recycling $NO_x$, as recently demonstrated by Ye et al. (2016b).

Despite the clear significance of atmospheric HONO, there is still considerable debate on its temporal and spatial distribution and its formation pathways. Observations typically show formation and accumulation of HONO in the nocturnal boundary layer, followed by a rapid decrease due to photolysis and enhanced vertical mixing in the morning (e.g. Wong et al., 2011). The nighttime formation of HONO is generally assumed to occur via the heterogeneous disproportionation of $NO_2$ in the presence of surface adsorbed water, with the following stoichiometry:

$$2NO_2 + H_2O \xrightarrow{surface} HONO + HNO_3 \tag{R1}$$

Nocturnal HONO formation is partly counteracted by HONO uptake on the ground, with reported surface uptake coefficients in the range of $2 \times 10^{-5} - 2 \times 10^{-4}$ (VandenBoer et al., 2013; Donaldson et al., 2014; Wong et al., 2012). Uptake on aerosol is also likely, but little information on this process is available.

During the day, the loss of HONO is dominated by its photolysis into OH and NO at wavelengths below 400 nm (Cox and Derwent, 1976):

$$HONO + h\nu \, (\lambda < 400nm) \rightarrow OH + NO \tag{R2}$$

This photolysis is, in part, balanced by the gas-phase reaction of OH and NO:

$$OH + NO \xrightarrow{M} HONO \tag{R3}$$

For many years, reactions (R2), (R3), and the less important reaction of HONO + OH, were assumed to be the only significant daytime HONO reactions. HONO concentrations were thus believed to be in a pseudo steady state (PSS) determined by its photolysis rate ($J_{HONO}$), [OH], and [$NO_2$]:

$$[HONO]_{SS} = \frac{k_{HONO}[NO][OH]}{J_{HONO} + k_{HONO+OH}[OH]} \tag{1}$$

Over the past decade, measurements have shown that daytime HONO mixing ratios tend to be 10 to 100 times higher than this PSS (e.g. Kleffmann et al., 2005; Kleffmann, 2007; Wong et al., 2012; Zhou et al., 2011). However, the precise chemical mechanism(s) responsible for the additional HONO in the atmosphere has yet to be clearly identified. A number of gas-phase, aerosol, and ground surface processes have been proposed to explain daytime HONO mixing ratios above its PSS.




Suggested daytime gas phase formation pathways of HONO (Bejan et al., 2006; Li et al., 2008, 2014) have been found to be unimportant in the atmosphere (e.g. Sörgel et al., 2011; Wong et al., 2012; Ye et al., 2015). Daytime HONO formation pathways on aerosol and ground surfaces include the photo-enhanced heterogeneous conversion of $NO_2$ to HONO on $TiO_2$ (Bedjanian and El Zein, 2012; Langridge et al., 2009; Ndour et al., 2008, 2009), soot (Ammann et al., 1998; Aubin and Abbatt, 2007; Gerecke et al., 1998; Khalizov et al., 2010; Monge et al., 2010), humic acid (Bartels-Rausch et al., 2010; Stemmler et al., 2006, 2007), and organic films (Brigante et al., 2008; George et al., 2005; Gutzwiller et al., 2002). While we know little about the detailed mechanism, Stemmler et al. (2006) proposed that the formation of HONO from $NO_2$ to HONO conversion on surface adsorbed humic acid is first order in $NO_2$ at low $NO_2$ mixing ratios, and linearly dependent on irradiance.

$$P_{HONO} \propto surface\,area \times [NO_2] \times irradiance \qquad (2)$$

Photolysis of surface adsorbed $HNO_3$ and aerosol $HNO_3$ is another proposed heterogeneous daytime source of HONO (Zhou et al., 2002, 2003, 2007; He et al., 2006; Beine et al., 2002; Dibb et al., 2002). The proposed mechanism is similar to that of aqueous nitrate photolysis (Mack and Bolton, 1999):

$$HNO_3(ads) + h\nu \rightarrow [HNO_3]^*(ads) \qquad (3)$$

$$[HNO_3]^*(ads) \rightarrow HNO_2(ads) + O(^3P)(ads) \qquad (4)$$

$$[HNO_3]^*(ads) \rightarrow NO_2(ads) + OH \qquad (5)$$

The linear relationship between HONO production rate and the product of particulate nitrates and its photolysis was recently confirmed by Ye et al. (2016b) in the marine boundary layer, where this mechanisms plays an important role in recycling $NO_x$. While there is still some debate about the absorption cross section of surface adsorbed $HNO_3$/nitrate, Ye et al. (2016a) report that the photolysis of $HNO_3$/nitrate is one to three orders of magnitude faster than that of gas-phase $HNO_3$, confirming that this process is a significant source of HONO in many environments.

Su et al. (2011) proposed that soil nitrite, through a reversible acid-base reaction of nitrite produced by biological nitrification and denitrification processes, can serve as a strong source of atmospheric HONO. Their chamber experiment shows that agricultural soils with low pH are particularly efficient sources of HONO. VandenBoer et al. (2015) proposed that HONO deposited on the surface during the night could act as a HONO source the following day, through the displacement of HONO from surface nitrate by atmospheric strong acids, such as HCl and $HNO_3$. Another proposed HONO source is the uptake of peroxynitric acid ($HO_2NO_2$) on the ground or aerosol surfaces that results in the formation of surface nitrite, which then reacts with hydrogen ions to produce HONO (Zhu et al., 1993). Lastly, HONO can also be directly emitted by combustion sources, such as vehicles, power plants, and also biomass burning (Kirchstetter et al., 1996; Neuman et al., 2016)

The multitude of proposed HONO formation mechanisms motivates additional field observations to shed light on the significance of the various processes and to elucidate the question of the nature of the reactive surface, i.e. ground or aerosol. In





addition, advances in our understanding of HONO formation requires observations that allow the analysis of the HONO budget, the HONO formation rate, and the identification and quantification of specific sources. Observations are also somewhat skewed to urban environments and summertime periods, despite the fact that the significance of HONO is likely higher in winter.

Here we address some of these questions by presenting observations of HONO and HONO precursor concentrations, as well as vertical fluxes, at a polluted wintertime rural location in the Uintah Basin (UB), Utah, USA. The environment is substantially different than that of most other observations in that temperatures were below freezing during most of the experiment, but little to no snow was present. The overarching goal of our observations was to determine whether HONO surface fluxes are able to close the daytime HONO budget, i.e. if in this specific environment the missing HONO source is the ground, and to investigate

whether any of the HONO formation mechanisms proposed in literature dominate the HONO formation.

## 2   Experimental

The 2012 Uintah Basin Wintertime Ozone Study (UBWOS) took place between January and early March 2012 in the Uintah Basin, Utah, USA, to study elevated wintertime $O_3$ formation (e.g. Edwards et al., 2014). The UB is located in the northeast corner of Utah and is part of the Colorado Plateau. The floor of the UB is about 1570 m above sea level (Lyman and Shorthill,

2013; Stoeckenius, 2015). The basin is mostly rural, with a population of about 50000, mainly living in three towns: Duchesne, Roosevelt, and Vernal. The economy of the UB is primarily based on energy production from petroleum resources, with approximately 10000 active oil and gas wells. The Bonanza 500 MW coal-fired power plant is located in the UB, about 20 km southeast of the UBWOS main site.

   The measurement main site, also known as Horse-pool site $(40°8'37.339", 109°28'1.849"W)$, was located in the northern

part of the UB, about 35 km south of Vernal. A variety of measurements were performed at the site, including observations of basic meteorological and radiometric parameters, as well as measurements of different trace gas concentrations. Here we will focus on the observations of concentrations and concentration gradients/fluxes made by UCLA's Long-Path Differential Optical Absorption Spectroscopy (LP-DOAS) instrument (Stutz and Platt, 1997; Wong et al., 2011, 2012; Tsai et al., 2014).

### 2.1   LP-DOAS Instrument

The LP-DOAS is an active DOAS system which consists of a Newtonian telescope that sends and receives a collimated light beam emitted by a high-pressure Xe-arc lamp. The light beam is folded once by an array of quartz corner cube retro-reflectors mounted at distances from 0.5 - 10 km. The light returned by theses retro-reflector arrays is fed into a spectrometer via a 350 $\mu$m quartz fiber that is part of a mode mixer (Stutz and Platt, 1997). Atmospheric absorption spectra were recorded at a resolution of about 0.5 nm by a Czerny-Turner type spectrometer with a focal length of 0.5 m (Acton Spectra Pro 500) in the

wavelength range of 300-380 nm. A Princeton Instrument Pixis 256 CCD camera was used for the detection of the spectra. The multichannel scanning technique (MCST) was employed to overcome pixel-to-pixel sensitivity variations (Brauers et al.,



1995; Platt and Stutz, 2008). Details on the instrument and the MCST can be found in Wong et al. (2012), Stutz et al. (2002), and Alicke et al. (2003).

The LP-DOAS system measured mixing ratios of $NO_2$, $O_3$, HONO, HCHO, and $SO_2$ at the Horse-pool site from January 22 to February 28, 2012. Here we will focus on the $NO_2$, $O_3$, and HONO data. The LP-DOAS telescope was located 2 m above the ground, and three retro-reflector arrays were installed at the base, midway, and top of a bluff located about 2.5 km northwest of the main site (Figure 1), at heights of 31 m, 45 m, and 68 m above the height of the LP-DOAS telescope. Figure 1 shows the location of the retro-reflectors relative to the location of the instrument and the three light paths (lower, middle, and upper). The retro-reflectors arrays were installed horizontally close to each other (within ∼64 m). Surface topography, determined from online maps, is very similar under each light path (Figure 1a). The light paths cross frozen natural soil and dry wintertime vegetation.

The LP-DOAS sequentially aimed at the three reflector arrays with a repetition time of 20-45 min, depending on visibility. The observations were continuous, except for periods with heavy fog and times at which instrument maintenance was performed. A measurement on one retro-reflector consisted of a sequence of an atmospheric absorption spectrum, a background spectrum measurement with the lamp blocked and, at low $NO_2$ mixing ratios, an additional Hg reference spectrum. This sequence was repeated nine times, each with a wavelength offset of 0.8 nm. Each LP-DOAS absorption spectra underwent correction for electronic offset and atmospheric background light. The MCST procedure (Alicke et al., 2002; Platt and Stutz, 2008) was used to construct the final absorption spectrum from one sequence of nine different wavelengths. The MCST composite spectrum then underwent a low-pass filter using a 20-fold triangular smoothing (Platt and Stutz, 2008).

Trace gas reference spectra were calculated from literature absorption cross sections (Table 1) after convolution with the instrument function, which was determined by measuring the 334 nm mercury atomic emission line, and a MCST simulation with the same wavelength shift parameters as the atmospheric spectrum (Platt and Stutz, 2008). A linear combination of trace gas reference spectra, together with a polynomial function and a spectrum of the Xe-Arc lamp, were fitted to the atmospheric spectrum using a combination of a linear and non-linear least-squares fit (Stutz and Platt, 1996; Platt and Stutz, 2008). A small (< 1 pixel) joint spectral shift was allowed for the trace gases in the fit. The fit was performed in wavelength intervals specific for each trace gas (see Table 1) to optimize the detection limits and stability of the fit. Table 1 lists the references included in each fit window and the average detection limit throughout the experiment. Figure 2 shows the results of a spectral analysis of $NO_2$ and HONO for an atmospheric spectrum taken during UBWOS 2012.

The analysis procedure also calculates mixing ratio errors for each spectrum. We will use these $1\sigma$ errors throughout the manuscript and propagate them for each data point in all calculations. In addition, the cross sections have an error that varies between 2% to 6% (see errors in parentheses in Table 1). The systematic error of the DOAS spectrometer is less than 3% (Stutz and Platt, 1996). The average detection limit for each retrieved trace gas was calculated by doubling the average error of each trace gas (Table 1).





## 2.2 Ancillary measurements

Extensive chemical and meteorological measurements were made at the Horse-pool site. This section describes the specific measurements that were used in this study for the data interpretation and modeling efforts. A summary of all the measurements made during UBWOS 2012 can be found in Lyman and Shorthill (2013).

### 2.2.1 Meteorological and micrometeorological observations

Temperature, relative humidity, wind direction, and wind speed were measured at the top (about 19 m agl) and 13.5 m agl of a sampling tower. Temperature and relative humidity were measured with a Vaisala model HMP45AC and pressure was measured with a Vaisala model PTB101B at 13.5 m agl. Wind direction and wind speed were measured with a vane anemometer (RM Young, model 05103) at 19 m agl, and a 3-dimensional sonic anemometer (RM Young model 81000) at 13.5 m agl. Data are reported as one-minute averages and the uncertainty is ±3%. Turbulent heat flux, Monin-Obukhov length, and friction velocity were calculated using micrometeorological data collected by a second sonic anemometer (Applied Technologies, Inc) installed on a separate tower at 16 m agl. Persson et al. (2002) estimated a random error of ±3 W m$^{-2}$ for turbulent heat flux and ±0.05 m s$^{-1}$ for friction velocity.

### 2.2.2 NO$_x$ and HNO$_3$ measurements

NO$_x$ measurements were made with a two channel O$_3$-NO chemiluminescence detector (Williams et al.). One of the channels determined NO and the other measured NO plus a fraction of NO$_2$ that had been converted to NO by a photolytic converter. The total uncertainty of NO was determined to be 4%, with a detection limit of 6 ppt (one minute average).

Nitric acid (HNO$_3$) was measured by negative ion proton transfer chemical ionization mass spectrometry (NI-PT-CIMS), with acetate as the reagent ion. This method is described in detail by (Roberts et al., 2010) and (Veres et al., 2008). The NI-PT-CIMS was placed on the tower at 10 m agl. The uncertainties for both HNO$_3$ was ±20%.

### 2.2.3 Photolysis rates

Photolysis rates of NO$_2$ and O$_3$ (O$^1D$ channel) were measured by commercial filter radiometers (Metcon, Inc.) mounted at 10 m agl. The rates were measured at a frequency of 1 Hz from 15 January until 22 February, until strong wind gusts toppled the tower that held the instrument. Photolysis rates of HONO were calculated based on filter radiometer data (Stark et al., 2007). Total uncertainties for one-minute average data are ±15% for J$_{NO2}$, ± 25% for J$_{O(1D)}$, and ±14% for J$_{HONO}$

### 2.2.4 VOCs measurements

A broad set of volatile organic compounds (VOCs) were measured in-situ by a gas chromatograph-mass spectrometer instrument (GC-MS). The inlet of the GC-MS was set up at approximately 20 m agl. A total of 65 VOCs were measured including alkanes, alkenes, cycloalkanes, aromatics, oxygenated VOCs, and nitrogen and halogen containing species. Uncertainties were



about 15-25% for hydrocarbons and 20-25% for oxygenates compounds. A detailed description of the instrument is described in the work of Gilman et al. (2010) and Goldan et al. (2004).

### 2.2.5   Aerosol number size distribution

Aerosol number size distribution was measured with a Scanning Mobility Particle Sizer (SMPS, TSI 3080 coupled to a TSI

3010 CN counter) and an Aerodynamic Particle Sizer (APS, TSI 3321). Aerosol particles were sampled at 12 m agl. The SMPS counted particles ranging from 20 nm to 500 nm geometric diameter; the APS collected particles into 34 size bins with aerodynamic diameters between 0.7 and 10.37 micrometers. The number size distributions were collected every 5 minutes.

### 2.3   Chemical Models

To aid in the interpretation of the UBWOS 2012 observations two chemistry and transport models were used. A one dimension

chemical and transport model (RCAT 8.2, run for this study) modeled vertical HONO concentration profiles and fluxes for direct comparison with the observations, while the OH output from the Weather Research and Forecasting model coupled with chemistry was used in the HONO budget calculations. The following sections will briefly describe both models.

### 2.3.1   RCAT

RCAT8.2 is a highly resolved one-dimensional chemistry and transport model that has been used to investigate the impact of

nocturnal processes and daytime chemistry of HONO and $NO_3$ (e.g. Wong and Stutz, 2010; Wong et al., 2013; Tsai et al., 2014). It is based on the gas-phase Regional Atmospheric Mechanism (RACM) which contains 77 species and 237 reactions, and aggregates atmospheric VOCs into 23 classes: 4 alkanes, 4 alkenes, 3 biogenic, 3 aromatics, and 9 carbonyls (Stockwell et al., 1997). RCAT8.2 also includes biogenic and anthropogenic emissions and the Mainz Isoprene Mechanism (MIM) (Geiger et al., 2003). Aerosol uptake rates in RCAT 8.2 are calculated using the mass-transfer equation by (Fuks and Sutugin, 1971).

The uptake rates were calculated based on the vertical profile of the aerosol surface to air volume ratio and a particle diameter of 150 nm. A unique feature of RCAT is that heterogeneous chemistry on the ground is modeled through direct molecular interactions. The vertical grid subdivides the lowest 2000 m of the atmosphere into 23 boxes with a logarithmic spacing below 1 m to account for the inefficient vertical transport near the ground (grid points: $1 \times 10^{-4}$, $1 \times 10^{-3}$, $1 \times 10^{2}$, 0.1, 1, 3, 6, 10, 20, 33, 50, 78, 90, 110, 121, 150, 175, 255, 300, 556, 750, 1000 and 2000 m). Vertical transport was constrained by

observed meteorology, with a predominately neutrally stable atmosphere during the day. The canopy height was set to 3 m; the displacement height was set to 4.27 m, and the surface roughness length was 0.03 m, to account for vegetation and small scale topography. The chemistry portion of the model was initialized at sunset with NOAA in-situ measurements of: hydrocarbons (HCs) (measured by GC-MS), CO (measured by CO instrument), $CH_4$ (measured by the Picarro $CO_2$/$CH_4$ instrument), PAN (measured by PAN GC), and UCLA LP-DOAS measurements of $NO_2$, $O_3$, HCHO, HONO, and $SO_2$. The initial concentrations

of the trace gases changed depending on the modeled day, with sunny day averages at sunset of 2320 ppb for $CH_4$, 92.7 ppb for alkanes, 4.1 ppb for aromatic VOCs, 1.6 ppb for carbonyls, 0.45 ppb for PAN, and negligible biogenic VOC. Hourly NO





emissions were adjusted to reproduce the LP-DOAS $NO_2$ observations. The aerosol surface vertical profiles were assumed to be constant up to the PBL height and decreased exponentially, to one-third of its surface concentration, above the PBL. The hourly aerosol surface area density diurnal values were calculated using Scanning Mobility Particle Sizer (SMPS) and Aerodynamic Particle Sizer (APS) measurements. Soil NO emissions, $\alpha$-pinene and isoprene emissions were assumed to be

$1\text{x}10^{10}$ molec. $\text{cm}^{-2}$ $\text{s}^{-1}$, $1\text{x}10^6$ and $1\text{x}10^8$ molec. $\text{cm}^{-3}$ $\text{s}^{-1}$ respectively. The aerosol and ground uptake coefficients adopted in RCAT 8.2 were $2\text{x}10^{-3}$ for $NO_3$, $2\text{x}10^{-5}$ for $NO_2$, $10^{-4}$ for HONO, 0.1 for $HO_2$, and $5\text{x}10^{-5}$ for $O_3$.

### 2.3.2    Weather Research and Forecasting model coupled with Chemistry (WRF-Chem)

A recent version of the WRF-Chem model (version 3.5.1 http://ruc.noaa.gov/wrf/WG11/) was used to simulate $O_3$ levels during UBWOS 2012 (Ahmadov et al., 2015). Gas-phase chemistry was based on the RACM mechanism (Stockwell et al., 1997), and

the photolysis rates were simulated using the Tropospheric Ultraviolet and Visible (TUV) photolysis scheme (Madronich, 1987). The initial meteorological and boundary conditions were based on the North American Mesoscale analysis fields (www.emc.ncep.noaa.gov). The OH mixing ratios and planetary boundary heights (PBLHs) model outputs were used in this study. A detailed description of the model results for UBWOS 2012 can be found in (Ahmadov et al., 2015).

### 2.4    Analysis of the vertical distribution of HONO

LP-DOAS instruments have been used to retrieve deposition velocities, vertical gradients, and fluxes of $SO_2$ (Platt and Perner, 1979) and reactive nitrogen species (Stutz et al., 2004; Wang et al., 2006; Wong et al., 2012, 2013; Tsai et al., 2014). The conversion of horizontally and vertically averaged trace gas concentrations on multiple absorption light paths to vertical trace gas gradients is typically achieved using a geometric approach, where the ground is considered flat. While the setup of the LP-DOAS system is similar to that of previous studies, the terrain under the light paths during UBWOS exhibited signifi-

cant topographical variations compared to the height interval covered by the light paths (Figure 1). This setup thus makes a geometric approach challenging, and a better method to retrieve HONO vertical gradients was therefore developed.

### 2.4.1    Retrieval of HONO vertical gradients

Our retrieval approach is based on the assumption that the vertical HONO concentration profile (C(z)) is a function of altitude above the surface (z) for each point along the light path. From published field observations (VandenBoer et al., 2013; Zhang

et al., 2009) as well as model results (Wong et al., 2013) we know that HONO concentrations typically decay from the surface upwards in a non-linear way. This decay is also supported by the negative gradients in the path-averaged HONO concentrations we observed during UBWOS 2012 (see below). While a direct functional dependence of [HONO] with altitude cannot be established, an exponential decay with altitude seems to appropriately describe the profile. This is particularly true for the model results of Wong et al. (2013). We will show later that an exponential decay shape also agrees well with 1D modeling

studies for UBWOS. We therefore parameterized the HONO concentration profile using the following exponential function:

$$C(z) = C_0 + C_1 \cdot exp(-\frac{z}{C_2}) \tag{6}$$



where, $C_0$, $C_1$, and $C_2$ represent the HONO background concentrations, HONO surface concentrations, and a scale height, respectively. To simulate the path averaged concentration observed by the LP-DOAS, the light path of each reflector with length, $L_i$, was subdivided in 50 m intervals. For each interval, j, the height of the light path above ground, $z_j$, was determined based on the topography shown in Figure 1 and the light path geometry. Considering the slanted path with an elevation angle, $\alpha_i$, the path averaged concentration is then calculated:

$$M(HONO)_i = \frac{1}{L_i} \sum_{j=1}^{L_i/50m} \frac{50m}{cos(\alpha_i)} C(z_j) \tag{7}$$

To determine the factors $C_0$, $C_1$, and $C_2$ from the observed path averaged concentrations ($M(HONO)_i$) on each light path, a nonlinear least squares fit was performed, minimizing $\chi^2$:

$$\chi^2 = \sum_i (M(HONO)_i - [HONO]_i)^2 \tag{8}$$

where i is the light path index (lower, middle, and upper light path). The trust-region-reflective algorithm, which is based on the interior-reflective Newton method described in Coleman and Li (1996), was used in this optimization (Matlab, lscurvefit). A lower allowable bound of 0 for both $C_0$ (HONO background) and $C_1$ (HONO surface concentrations) were specified in the fit. The decay term $C_2$ was left unbound (varying from –infinity to +infinity). The fit thus allows for decreasing, constant, or increasing altitude profiles, as long as the concentrations remain positive. Figure 3 shows the results of such a retrieval for February 7, 2012, at 11:00, 12:00, and 13:00 LT. The comparison with results from RCAT 1D model for these specific times confirms that Equation 6 describes the expected HONO concentration profile well. The slight differences between the retrieved and model profiles at 11:00 and 12:00 LT can be explained by the uncertainty in the vertical mixing parameters used in the model, i.e. stronger vertical mixing in the model leads to modeled profiles which are less steep than the one retrieved from the observations. The modeled and retrieved profiles at 13:00 LT, on the other hand, show an excellent agreement, implying that the combination of surface source and vertical mixing in the model match those during the observations well.

### 2.4.2 Retrieval of HONO surface fluxes

The gradient method (Businger, 1986) was used to calculate HONO surface fluxes, i.e. the HONO flux is the product of the vertical concentration gradient, $\frac{d\bar{c}}{dz}$, and the eddy diffusivity coefficient ($K_c$):

$$F_c(z) = -K_c(z) \frac{d\bar{c}}{dz} \tag{9}$$

where the eddy diffusivity $K_c(z)$ at height (z) can be calculated using the Von Kaman constant (k = 0.37) (Telford and Businger, 1986), Monin Obhukov length (L), friction velocity (u*), displacement height (D), and the dimensionless stability correction factor ($\Theta(\frac{z}{L})$), which is a function of stability:

$$K_c = \frac{k \cdot u^* \cdot (z - D)}{\Theta(\frac{z}{L})} \tag{10}$$





$$\Theta(\frac{z}{L}) = \left(1 - \frac{16 \cdot (z - D)}{L}\right)^{-\frac{1}{2}} \qquad for \frac{z}{L} < 0 \quad (unstable) \tag{11}$$

$$\Theta(\frac{z}{L}) = \left(1 + \frac{5 \cdot (z - D)}{L}\right) \qquad for \frac{z}{L} > 0 \quad (stable) \tag{12}$$

$u^*$ and L were determined from the sonic anemometer observations described in section 2.2.1. The displacement height for UBWOS was determined using the zero-plane displacement height equation by Stanhill (1969):

$$log(D) = 0.9793 \times log(h) - 0.1536 \tag{13}$$

where h is the vegetation height. The displacement height for UBWOS, D=(0.39±0.17m), was calculated based on the average of eleven popular vegetation species in the Southern UB (mainly shrubs and grass; Butler and England (1979)). $K_c(z)$ was calculated for z = 16 m, the altitude of the micrometeorological measurements. The uncertainty of $K_c(z)$ was determined by propagating the random error of friction velocity and Monin-Obukhov length from the observations and the uncertainty in D.

To determine the HONO gradient in Equation 9 the retrieved HONO concentration profile was averaged in two vertical height intervals. The lower interval extended from 2 m (height of the instrument agl) to 16 m agl (height of the anemometer agl), and the upper interval extended from 16 m to 40 m agl (maximum height difference between the upper light path and the ground). 16 m agl is also the geometric mean height of the intervals. The difference between the average HONO mixing ratios in each box was divided by the average heights of the intervals ($\bar{z}_{upper} = 28$m, $\bar{z}_{lower} = 9$m):

$$\frac{d\overline{HONO}}{dz} = \frac{\overline{HONO}_{upper} - \overline{HONO}_{lower}}{\bar{z}_{upper} - \bar{z}_{lower}} \tag{14}$$

The uncertainty of the HONO gradient was determined by propagating the uncertainty associated with the average concentration of HONO in the lower and upper box. We assumed that the uncertainties in the boxes are equal to the HONO error of the DOAS analysis on the lower, middle, and upper light paths, weighted based on its contribution to the box height.

## 3 Results

### 3.1 Meteorology during UBWOS 2012

UBWOS 2012 was dominated by weak-wind periods, with average wind speeds ranging between 2.5 m s$^{-1}$ during the night to 4 m s$^{-1}$ in the evening (Figure 4b). Temperatures ranged between 259K and 286K, with diurnally average temperatures varying between 270 K in the early morning and 277 K in the mid afternoon (Figure 4a). Temperature measurements across the UB showed that daytime temperature inversions conducive to ozone formation were rare. Snow cover was almost absent during UBWOS 2012. Snow fell on 19 and 29 February, but thawed quickly due to the high temperatures. For most of the field study the ground consisted of frozen natural soil and dry vegetation.



### 3.2 NO$_2$, O$_3$, and HONO light-path average mixing ratios

Daytime and nighttime NO$_2$ mixing ratios were, on average, ~2 ppb and ranged from 0.10 $\pm$ 0.07 ppb to 28.8 $\pm$ 0.08 ppb on the lowest light-path (Figure 4c). While these high levels of NO$_2$ are comparable to NO$_2$ levels found in some polluted urban areas, their sources are different. In general, transportation is the main source of NO$_2$ pollution in urban areas but, in the UB, NO$_x$ was mainly emitted from oil and gas operations (57-61%) and to a lesser extent by emissions from the Bonanza power plant, although the power plant emissions often remain aloft and are not observed at the ground in the basin (Lyman and Shorthill, 2013).

O$_3$ mixing ratios ranged from 2.4 $\pm$1.7 to 60.4 $\pm$ 3.3 ppb, with daytime ozone maxima consistently below the U.S. National Air Quality Standard (75 ppb) (Figure 4d). O$_3$ levels measured in the UB in 2012 were lower than those observed in the subsequent two winters. The absence of snow likely led to more convective mixing which prevented pollutants from building up near the surface and reacting with sunlight to form O$_3$ (Lyman and Shorthill, 2013; Edwards et al., 2014). Furthermore, without snow cover, photochemical formation of O$_3$ was less efficient, as the actinic flux was 1.6 to 2 times smaller (Lyman and Shorthill, 2013; Edwards et al., 2014). HONO levels were also low compared to levels measured in polluted urban areas. The nighttime and daytime path-averaged HONO mixing ratios were 56 ppt and 78 ppt, respectively and ranged from below the detection limits to 274 $\pm$ 33 ppt (Figure 4e). The diurnally averaged mixing ratios of daytime HONO in the 2-70 m vertical height interval (or LP-DOAS upper height interval) ranged from 35 to 81 ppt, with an average of 48 ppt. These HONO levels are comparable to HONO observed in rural areas (e.g. Zhou et al., 2011) but lower than those observed in polluted urban environments (e.g. Su et al., 2008; VandenBoer et al., 2013; Wong et al., 2012).

### 3.3 Diurnal Profile during sunny days

To simplify the analysis of the behavior of HONO, its precursors, and the various environmental parameters impacting its chemistry, we will concentrate only on sunny days (27 January, 4 February, 6 February, 7 February, and 17 February) from here on, when the meteorological conditions were fairly constant and the relatively smooth actinic flux simplified the interpretation of the data. The restriction to sunny days also has the advantage that we can use the PSS for budget calculations due to the short photolytic lifetime of HONO (about 13 minutes near noontime and less than 21 minutes between 09:00 and 15:00 LT). Figure 5 shows the diurnal profile of J$_{HONO}$ (Figure 5a), NO$_2$ (Figure 5b), O$_3$ (Figure 5c), and HONO (Figure 5d) on each light-path (lower light-path in black, middle light-path in blue, upper light-path in red, and the 1$\sigma$ mixing ratio variation represented by the grey shadow) averaged over all sunny days.

The average diurnal profile of O$_3$ follows the expected behavior, with low mixing ratios of $\sim$30 ppb at night and increasing O$_3$ levels after sunrise, mostly due to the mixing of the PBL (Edwards et al., 2013). As the day progresses, O$_3$ levels reach a maximum of $\sim$46 ppb in the afternoon due to its photochemical formation. During the day, O$_3$ did not exhibit vertical gradients within our measurement uncertainty due to well mixed daytime atmospheric conditions. However, O$_3$ showed clear positive gradients at night. Together with the decreasing O$_3$ at night, this indicates that surface deposition likely played an important role (Sillman, 1999).



The sunny day diurnal average of $NO_2$ is higher at night, with maximum mixing ratios in the morning of ∼4 ppb and lower values later during the day of ∼1 ppb, when $NO_2$ undergoes photolysis and the boundary layer mixes. In contrast to $O_3$, a vertical gradient at night is less apparent, but a more detailed view (not shown here) reveals positive nocturnal gradients, consistent with surface deposition. Daytime gradients were not observed within the uncertainties of the observations.

HONO shows a diurnal profile similar to that reported from urban locations. It accumulates during the night with the largest hourly average HONO mixing ratio of 0.10 ppb at 06:00 LT. A distinctive decrease after sunrise due to its efficient photolysis and daytime mixing is observed, leading to minimum HONO levels of 0.03 ppb around 15:00 LT. Daytime HONO mixing ratios are smaller than those observed in polluted urban areas. For example, in Los Angeles the UCLA LP-DOAS measured on average about 100 ppt of daytime HONO on its lowest light-path (33-78 m agl). The hourly average diurnal profile of

HONO generally shows larger HONO mixing ratios in the lower than in the upper light-path, both during the day and night (Figure 5d). This negative HONO gradient indicates an upward flux and the presence of a surface source of HONO. The observation of daytime negative HONO gradients (Figure 3) are in agreement with HONO observations made in Boulder, Colorado (VandenBoer et al., 2013) where daytime mixing ratios were 100±80 pptv at the surface and 35 pptv aloft, and measurements in Houston, Texas (Wong et al., 2012) that also show larger HONO mixing ratios near the surface.

## 3.4   HONO Vertical Fluxes

Both the daytime vertical profiles and negative gradients of HONO (Figure 5d) point toward the existence of HONO surface fluxes. We therefore applied the methods outlined in Section 2.4 to calculate vertical HONO fluxes on sunny days. Figure 6 shows the components of such a calculations for 6 February, 2012, along with $NO_2$ mixing ratios measured on the lower light-path (Figure 6a). HONO mixing ratios in the lower (black markers) and upper (red markers) boxes calculated from retrieved

HONO vertical profiles (Figure 6b) and the gradient calculated using Equation 14 clearly show the negative gradient of HONO. By combining this gradient with the observed eddy diffusivity coefficients (Figure 6d) in Equation 9, the vertical HONO flux was determined. The relatively constant eddy diffusivity coefficients between 10:00 and 15:00 LT, along with the temporal trend of the HONO gradients, results in HONO surface fluxes that follow the solar irradiance temporal trend, with a maximum of $(4.9 \pm 0.2)$ x$10^{10}$ molec. cm$^{-2}$ s$^{-1}$ near noontime. It should be noted that the temporal variation of the flux seems to be

controlled by that of the HONO gradient. The source of the higher-frequency variation, i.e. at 13:30 LT, is unclear, but one can suspect that advection could have played a role. Whether the apparent anti-correlation of the HONO gradient and flux with $NO_2$ after 11:00 LT is meaningful will be discussed further below. It should also be noted that the turbulent mixing time scale at 16 m altitude is around 160 sec and is thus shorter than the photolytic lifetime of HONO. We thus report the flux without specifically considering HONO photolysis.

To allow a more thorough interpretation of our HONO flux calculations we averaged the hourly flux data over all sunny days of the experiment. The hourly average sunny day HONO flux (Figure 7) approximately follows solar irradiance, with a maximum of $(1.7 \pm 0.3) \times 10^{10} \mathrm{molec.cm^{-2}s^{-1}}$ at noontime. This value is comparable to the average HONO flux (about $1.15 \times 10^{10} \mathrm{molec.cm^{-2}s^{-1}}$) measured above a northern Michigan forest canopy (Zhang et al., 2009; Zhou et al., 2011), and lower than average noontime HONO flux ($3.01 \times 10^{10} \mathrm{molec.cm^{-2}s^{-1}}$) measured over an agricultural field in Bakersfield (Ren



et al., 2011). The upward HONO fluxes observed during sunny days implies that daytime HONO formation occurs at the ground and is driven by solar irradiance, as we will discuss further below.

### 3.5 PSS Daytime HONO budget

Similar to previous field observations, daytime HONO levels during UBWOS exceeded that of its photochemical steady state

levels (Eq. 1). The difference between the calculated $[\text{HONO}]_{\text{ss}}$ (Figure 8, red line) and HONO measured in the lowest light path (Figure 8, black line) points towards the presence of an unknown HONO source during the day. To investigate this missing source and to determine if the upwards HONO flux from the surface explains the missing source, we performed a HONO budget analysis using a PSS approach, similar to that used in (Wong et al., 2012). The short lifetime of HONO justifies the use of the PSS approach (Equation 15). It is assumed that HONO reaches a pseudo steady state between the known (Equations 16) and

unknown ($\text{P}_{\text{unknown}}$) sources of HONO and loss rates of HONO (Equation 17 and Equation 18). Direct emissions of HONO are not considered in these studies because our measurements are not significantly influenced by traffic.

$$\frac{d[HONO]}{dt} = P_{unknown} + HONO_{formation} - HONO_{photolysis} - HONO_{oxidation} = 0 \tag{15}$$

$$HONO_{formation} = k_{NO+OH}[NO][OH] \quad with \quad k_{NO+OH} = (1.8 \pm 0.4)10^{-13} molec.^{-1}cm^3 s^{-1} (275K, 843hPa) \tag{16}$$

$$HONO_{photolysis} = J_{HONO}[HONO] \tag{17}$$

$$HONO_{oxidation} = k_{HONO+OH}[HONO][OH] \quad with \quad k_{HONO+OH} = (4.3 \pm 0.8)10^{-12} molec.^{-1}cm^3 s^{-1} (275K) \tag{18}$$

The HONO formation and loss rates were calculated using in-situ NO measurements, $J_{\text{HONO}}$, LP-DOAS lower light-path

HONO mixing ratios, and OH concentrations modeled by WRF-Chem. Modeled OH mixing ratios were used due to the lack of OH in-situ measurements. An uncertainty of 20% was assigned to the WRF-Chem OH concentrations, which was determined by comparing the difference between OH calculated by RCAT 8.2 and WRF-Chem during sunny days. The HONO rate constants and their uncertainties were determined using data from Sander et al. (2011) and the uncertainty in temperature. Following Equation 15, the HONO formation rate needed to balance the budget, $\text{P}_{\text{unknown}}$, was calculated using:

$$P_{unknown} = J_{HONO}[HONO] + k_{HONO+OH}[HONO][OH] - k_{NO+OH}[NO][OH] \tag{19}$$

The uncertainties of the formation and loss rates of HONO and $\text{P}_{\text{unknown}}$ were calculated by propagating the uncertainties of their rate constants and measurement errors. Figure 8 shows the sunny day hourly average of $\text{HONO}_{\text{formation}}$ (red line),





HONO$_\text{photolysis}$ (green line), HONO$_\text{oxidation}$ (blue line), and P$_\text{unknown}$ (black line). Surface fluxes of HONO were converted into column HONO formation rates (Figure 8, magenta line) by dividing them by the height (H) at which the influence of HONO surface fluxes on the total HONO column becomes negligible. This height was determined from two RCAT 8.2 model runs for 27 January, one with and one without HONO surface fluxes. For each hour, an H was set at the height where the

difference between the HONO modeled with and without HONO surface fluxes was less than 1 ppt. The average H between 09:00 and 15:00 hours ($273\pm113$m) was used for the calculation of the HONO surface flux rate. The uncertainty of H is the $1\sigma$ deviation of H from 09:00 to 15:00 hours. The uncertainty of HONO flux rate was calculated by propagating the uncertainty of HONO flux and H, which are the main uncertainties in this calculation.

The comparison of the various terms in Equation 15 are shown in Figure 8, where the grey bars indicate morning and evening

periods when the HONO lifetime is too long to allow a PSS interpretation. The dominant HONO loss process is, as expected, photolysis. The HONO gas-phase formation from OH + NO does not fully balance this loss, thus clearly showing the need for an additional HONO source to close the budget. The comparison of this missing HONO source, P$_\text{unknown}$, with the rate from surface emissions shows that HONO formation on the surface can explain this missing source, within the uncertainties of the calculation, between 11:00 - 13:00 hours and is somewhat lower at 10:00 hours, when the photolysis of HONO formed

at night may still contribute a small amount of HONO. The fact that surface formation cannot fully explain P$_\text{unknown}$ at 14:00 hours is less clear, but may be a consequence of an increase of P$_{unknown}$ due to a decrease of P$_{NO+OH}$ caused by lower NO observations and lower modeled OH in the late afternoon.

### 3.6   Model Daytime HONO budget

We used our 1D chemistry and transport model, RCAT 8.2, to verify that HONO surface fluxes can account for the HONO

missing daytime source during UBWOS 2012. Initially, the model was run with its basic set up, which included: NO$_\text{x}$ emissions chosen to reproduce the observed NO$_2$ mixing ratios, HONO emissions calculated using a HONO/NOx = 0.008 emission ratio (Kurtenbach et al., 2001), gas phase formation of HONO from reaction of NO with OH, dark conversion of NO$_2$ on aerosol and ground surfaces (with a yield of 50% (Wong et al., 2011)), and HONO loss reactions which included HONO photolysis, reaction with OH, and uptake on aerosol and ground surfaces (with reactive uptake coefficients of $10^{-3}$ and $2 \times 10^{-5}$ respectively

(Wong et al., 2011)). It should be noted that the surface HONO formation from the dark conversion of NO$_2$ was negligible for the daytime runs. The comparison of HONO mixing ratios from the base model run and observations for two days of the experiment, February 4 and January 27 (Figure 9, blue line), confirms that gas-phase chemistry alone cannot explain the observed daytime HONO levels near noon time, when the photolytic source of HONO is largest. To test if the calculated HONO flux can explain this difference, a HONO surface flux term was implemented in the 1D model as a ground surface release. This

flux was determined from the calculated fluxes for the two days, scaled by a factor to account for the difference (caused by photolytic loss of HONO) between HONO fluxes at the surface and at the measurement height of 19 m agl. To determine the appropriate factor, we compared the measured and modeled HONO flux at 19 m agl in initial test runs, and then applied the factor for the final run.





As expected, the model run with the HONO flux set up (Figure 9, green line) produces more HONO than the basic model run (Figure 9, blue line). Within the uncertainties of the observations, the model with additional HONO flux reproduces the observed HONO mixing ratios well on January 27. The model also reproduces HONO well after noon on February 4, but overestimates HONO between 09:00 and 11:00 LT, when a plume of $NO_2$ was encountered. We will discuss this event in

more detail below. The disagreement during the early morning of February 4 is likely due to challenges of modeling the early morning transition from a stable to a neutral boundary layer during a time of fast changing actinic fluxes. Despite these shortcomings, the comparison of modeled and observed HONO mixing ratios confirms the findings of the PSS analysis that the observed vertical HONO fluxes explain the missing daytime HONO source in this environment.

## 4   Discussion

The vertical HONO concentration profiles (Figure 3) and the derived vertical fluxes (Figure 6e) clearly show the presence of an important ground surface source of daytime HONO. The strong daytime variation of the sunny day HONO fluxes, with an average upward fluxes at solar noon of $1.5 \times 10^{10}$ molec. $cm^{-2}$ $s^{-1}$, implies a source that is driven by solar irradiance (Figure 7). The precise mechanism of this HONO source, however, deserves further analysis.

Two previous studies of HONO fluxes measured with Relaxed Eddy Accumulation (REA) systems reported analyses of

daytime HONO surface formation. Zhou et al. (2011) measured HONO fluxes at 11 m above a forest canopy in Michigan during Summer, 2008. Their observed HONO flux correlated positively ($R^2 = 0.69$) with the product of photolysis rate constant of $HNO_3$ on leaf surfaces and leaf surface nitrate loading, suggesting that the photolysis of $HNO_3$/nitrate on forest canopies is a significant daytime source of HONO to the lower atmosphere in low $NO_x$ rural environments. On the other hand, Ren et al. (2011) measured the HONO flux 18 m agl above an agricultural field in Bakersfield, California, during the spring of

2010. Their daytime HONO flux showed a strong correlation ($R^2 = 0.99$) with the product of $NO_2$ concentrations and solar radiation instead.

We performed a similar analysis based on our observation. Because positive HONO fluxes point to formation on the ground (rather than in the gas phase or on aerosols), we used solar irradiance instead of photolysis rates for our analysis (see Wong et al. (2012) for further explanation of this approach). Only visible solar radiance was measured during UBWOS, which can

be used to describe photo-enhanced $NO_2$ conversion; but because $HNO_3$/nitrate species absorb in the UV range and UV solar irradiance was not measured in this study, its value was estimated by scaling visible solar irradiance with the measured $J_{HNO_3}$ and $J_{NO_2}$ ($UVSolarRad = SolarRad \times (J_{HNO_3}/J_{NO_2})$). We used gas-phase $NO_2$ and $HNO_3$ mixing ratios as proxies for surface adsorbed $NO_2$ and $HNO_3$, i.e. we assumed that gas-phase and surface adsorbed concentrations were in a equilibrium.

Comparison of our calculated HONO flux, with the product of $NO_2$ mixing ratios and visible solar irradiance, and the prod-

uct of gas phase $HNO_3$ mixing ratios and UV solar irradiance (Figure 10) show that they have a similar variation pattern, and generally follow the same diurnal variation trend as solar irradiance. Daytime HONO flux strongly correlates with both $[NO_2]$ $\times$ solar radiation ($R^2 = 0.58$) (Figure 11a) and $[HNO_3] \times$ UV solar radiation ($R^2 = 0.66$) (Figure 11b). The correlation coef-



ficient is slightly larger for the [HNO$_3$] × UV solar radiation, but the difference is small enough to make a clear determination of the dominant pathway difficult.

A large plume event on 4 February at the Horse-pool site provided an opportunity to examine the photo-enhanced NO$_2$ conversion at the ground as a potential HONO formation pathway. Comparison between NO$_2$ and SO$_2$ mixing ratios measured
in the lower light path (not shown) shows tight correlation (R$^2$ = 0.99) between the two species, suggesting that the source of the plume emitted both of these trace gases. Further analysis shows that the plume had not aged significantly, as illustrated by the missing peak of HNO$_3$. In addition, aircraft measurements (made by NOAA's Global Monitoring Division and not shown here) show CO levels consistent with levels observed in coal-fired power plants. Hence we concluded that the plume originated at the Bonanza power plant (Lyman and Shorthill, 2013).

We observed a peak of HONO mixing ratios and HONO flux (Figure 9c,e) that coincided with the plume of NO$_2$ around 10:00 hours (Figure 9a). The correlation of HONO mixing ratios, HONO flux, and NO$_2$ mixing ratios, and the absence of a gas-phase HNO$_3$ peak at this time, suggest that the source of the HONO peak was from the ground and that it involved NO$_2$ molecules.

To further confirm this interpretation we used the RCAT model to simulate this specific event. The model was constrained by
meteorological and radiative observations, and initialized with the trace gas observations as described in Section 2.3.1 (Figure 9a). The model also included HONO formation from photolytic conversion of NO$_2$ on the ground, which was parameterized by a solar irradiance dependent reactive uptake coefficient. The solar irradiance was described by a cubic dependence on J$_{NO2}$ (Wong et al., 2013), and the noontime reactive uptake coefficient was adjusted to a somewhat smaller value ($1.4 \times 10^{-5}$) than the one used in Wong et al. (2013), as the value used in their study lead to an overprediction of HONO. A lower coefficient can
likely be explained by the very different environments between our study, i.e. low soil temperature (in this study) compared to high temperatures in the urban environment of Houston (Wong et al., 2013).

Taking into account the hourly results of our model, the model simulations driven by the observed fluxes as well as the photo-enhanced NO$_2$ conversion capture the HONO mixing ratio peak between 10:00 and 11:00 LT. The agreement between the prescribed and model flux also supports the conclusion that photo-enhanced NO$_2$ conversion on the surface explains HONO
formation in this plume.

We then applied the same model to the observations on 27 January 2012, a day characterized by average wind speeds and NO$_2$ mixing ratios around 2 ppb (Figure 9). We again adapted the emissions to simulate the general NO$_2$ mixing ratios. The model overpredicts NO$_2$ in the morning, but captures the observations well after 11:00 LT. Despite the high NO$_2$, the base run (blue) and photo-enhanced run (red) underestimate measured HONO mixing ratios (black), suggesting that on days of
moderate NO$_2$ levels, photo-enhanced conversion of NO$_2$ does not seem to play an important role in HONO production. The strong correlation between HONO flux and [HNO$_3$] × UV solar radiation suggest that photolysis of surface HNO$_3$/nitrate might explain the observed HONO flux. However, without further information, such as concentration of surface adsorbed HNO$_3$, is not possible to unequivocally determine whether this mechanism was the source for the vertical HONO flux on 27 January.





An alternative HONO formation mechanism is the daytime displacement of HONO by strong acids. VandenBoer et al. (2014) showed evidence of uptake of HONO on soil surface at night and subsequent HONO displacement from soil by deposition of strong acids, such as $HNO_3$ and HCl, during the day. They argue that HONO formed through the acid displacement mechanism is proportional to the product of solar radiation and $NO_2$ concentration, as production of $HNO_3$ is proportional to the concentration of $NO_2$ and OH, the latter of which is proportional to solar radiation. For this mechanism to contribute significantly to the next day daytime HONO surface flux, it requires a substantial deposition of HONO during the previous night. To determine whether this mechanism played an important role in UBWOS 2012, we calculated the total number of HONO molecules deposited during the night of 27 January using the nighttime model results, and compared it to the total number of HONO molecules released by the daytime HONO surface flux. The integrated surface HONO concentration from 19:00 of the previous night to 07:00 LT was about $1.84 \times 10^{14}$ molec. $cm^{-2}$. This number is lower than the integrated daytime HONO flux, calculated from the net HONO flux from 07:00 to 17:00 LT, of $2.84 \times 10^{14}$ molec. $cm^{-2}$. If every HONO molecule deposited at night would be re-released, this mechanism would be able to explain up to 65% of the vertical flux. However, in this case one would expect that, as the surface reservoir is depleted throughout the night, the flux would also decrease, which we did not observe. Nevertheless, we cannot rule out that the re-release of surface deposited HONO contributes to the overall vertical HONO flux during the day.

Another potential HONO formation mechanism is its formation from soil nitrite (Su et al., 2011). Su et al. (2011) showed that gas-phase HONO released by soil depended strongly on the concentration of nitrite in soil and its pH. They proposed that biological nitrification and denitrification processes are the main sources of nitrite ions in the soil, and showed that fertilized soils with low pH appeared to be particularly strong sources of HONO. We believe that emission of HONO from soil nitrite played a minor role in UBWOS 2012, as the soil was alkaline and cold during this field study. Measurements by NOAA measured a pH of 8.06 for a soil sample collected near the Horse-pool site in 2014 (pers. comm) and a report by Utah State University Extension stated that the pH of a large portion of soils in Utah ranged between 7.8 and 8.2 (https://extension.usu.edu/files/publications/publication/AG-SO-07.pdf). Like most biological reactions, nitrification and denitrification processes are strongly influenced by soil temperatures, with these processes ceasing at soil temperatures below $5°C$ (Western Plant Health Association, 2002). The hourly average soil temperatures at noon in 2014 was about $-4°C$ (measured at 5 cm below the surface); therefore we can assume that soil temperatures in 2012, which were not directly measured, were not as cold as in 2014, but, nevertheless, cold enough to render nitrite production from biological reactions negligible.

Comparison of the missing source of HONO and HONO surface flux rate (Figure 8) shows that HONO surface flux rate (magenta line) accounts for a significant portion of the $P_{unknown}$ (63±32%) throughout the day (from 09:00 to 15:00 hours). Furthermore, near noontime (from 11:00 to 13:00 hours), surface flux rate accounts for an average of 90±30% of $P_{unknown}$, suggesting that photolytic surface sources of HONO, and not HONO formation pathways in the gas phase or on aerosols, were the dominant sources of the unknown HONO in 2012.

Under high NOx events, our HONO flux analysis favors HONO formation mechanisms that are related to solar radiation and $NO_2$ concentrations, such as photo-enhanced conversion of $NO_2$ and nighttime uptake of HONO followed by daytime acid displacement. Under moderate or low $NO_2$ level conditions, photolysis of $HNO_3$ seems to be the main HONO source.





The $P_{\text{unknown}}$ peak at 08:00 - 09:00 LT is probably the result of rapid photolysis of HONO that accumulated in the nocturnal boundary layer, and ineffective HONO production (through NO + OH) in the morning. Nitrophenols were not measured during UBWOS 2012, however, they were measured in 2014 in UB during the winter (Yuan et al., 2016). Yuan et al. (2016) determined that the main source of nitrophenols in UB in 2014 was the oxidation of aromatics and that primary emissions of nitrophonels

were negligible. Even though aromatics such as benzene were quite large in 2012 (median benzene in UB 2012 = 0.72 ppbv), photochemistry in 2012 was much less active compared to 2014 "suggesting that secondary formation of nitrophenols would be a relatively small source in 2012 and that the subsequent photolysis of nitrophenols would not be an important source of HONO."

## 5   Conclusions

HONO vertical concentration profiles and fluxes were measured during the winter of 2012 in the Uintah Basin, Utah using UCLA's LP-DOAS instrument. The UB is highly impacted by oil and gas activities and thus exhibits $NO_2$ mixing ratios similar to those of polluted urban and sub-urban areas. Snow was almost completely absent in 2012, thus allowing the study of HONO chemistry over cold natural soil.

HONO mixing ratios near the surface ranged from below detection limits to $270 \pm 20$ ppt, with a campaign average of

74 ppt. These levels are lower than HONO measured in polluted urban areas, but comparable to levels measured in rural environments. The sunny day hourly average HONO fluxes generally followed the diurnal variation of solar irradiance, with a noontime maximum of $(1.7 \pm 0.3) \times 10^{10} \text{molec.cm}^{-2}\text{s}^{-1}$. The upward fluxes supported the idea that HONO is formed by a photolytically driven surface source. A PSS and model HONO budget calculation clearly show that this surface source closes the daytime HONO budget, i.e. it is responsible for the commonly identified missing daytime HONO source.

The determination of the precise chemical mechanisms responsible for the surface flux is challenging due to the dominating correlation with solar irradiance of most of the mechanisms. However, our results seem to indicate that both photolysis of surface $HNO_3$/nitrate and photo-enhanced $NO_2$ conversion are likely active. Acid displacement of HONO deposited during the night, as proposed by (VandenBoer et al., 2014), can contribute to the observed HONO flux, but a calculation of the total deposited HONO throughout the night with the integrated daytime flux reveals that this mechanism by itself cannot explain

the observed daytime HONO formation. The low temperatures and alkaline soil make the bacterial production of nitrite and its release (Su et al., 2011) likely unimportant.

Our results illustrate the significance of heterogeneous chemistry on the ground in converting adsorbed reactive nitrogen species into HONO, thus impacting the OH radical and nitrogen budget in the surface layer. Our analysis points to the difficulty of clearly identifying the precise chemical reaction mechanism forming HONO due to their strong correlation with

solar irradiance and the cross-correlation of the involved nitrogen species. More focused experiments that include a better characterization of the chemical composition of the surface are needed to further investigate HONO chemistry.



*Acknowledgements.* Funding for the UCLA part of this study was provided by the National Science Foundation (award no. 1212666). The 2012 Uintah Basin Winter Ozone Study was a joint project led and coordinated by the Utah Department of Environmental Quality (UDEQ) and supported by the Uintah Impact Mitigation Special Service District (UIMSSD), the Bureau of Land Management (BLM), the Environmental Protection Agency (EPA) and Utah State University. This work was funded in part by the Western Energy Alliance, and

5  NOAA's Atmospheric Chemistry, Climate and Carbon Cycle program. We thank Questar Energy Products for site preparation and support.





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





**Table 1.** Wavelength windows, cross sections, and fitted trace gases used for DOAS retrieval during UBWOS 2012. The quoted detection limits are campaign averages.

| Trace gases | Wavelength Range (nm) | Cross Sections Rreference and Uncertainty | Fit Spectral References | Average Detection Limits |
|---|---|---|---|---|
| HCHO | 324-346 | (Meller and Moortgat, 2000), 280K, 5% | $NO_2$, HONO, HCHO, $O_3$, $O_4$ | 0.3 ppb |
| HONO | 338-344 & 351-371 | (Stutz et al., 2000), 298K, 5% | $NO_2$, HONO, HCHO, $O_3$, $O_4$ | 0.03 ppb |
| $NO_2$ | 338-344 & 351-371 | (Vandaele et al., 1998), 294K, 2% - 5%; (Voigt et al., 2002), 260K, 4% - 6% | $NO_2$, HONO, HCHO, $O_3$, $O_4$ | 0.07 ppb |
| $O_3$ | 324-346 | (Serdyuchenko et al., 2014), 273K, 3% | $NO_2$, HONO, HCHO, $O_3$, $O_4$ | 1.4ppb |
| $SO_2$ | 305-310 & 313-321 | (Vandaele et al., 1994), 296K, 2.4% | $NO_2$, HCHO, $O_3$, and $SO_2$ | 0.05ppb |

$O_4$ literature cross section: (Hermans et al., 1999), 298K





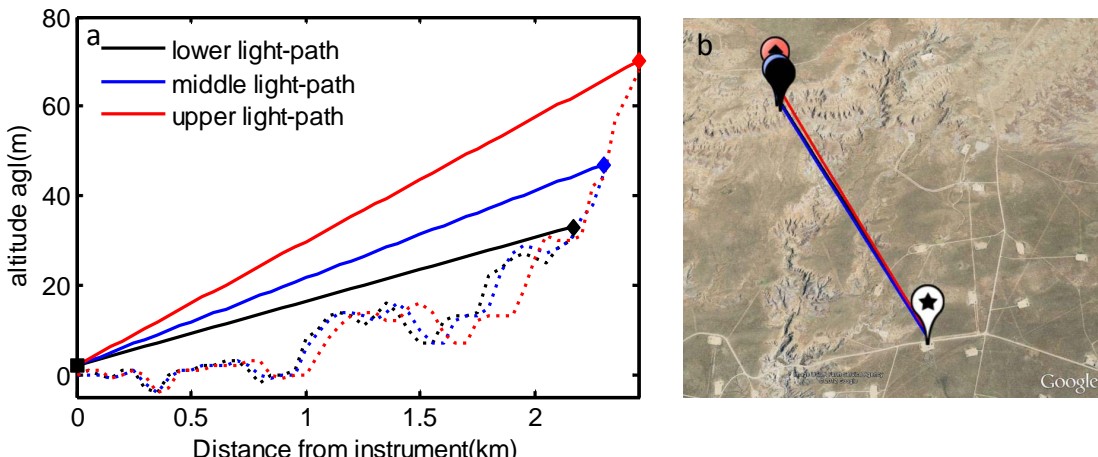

**Figure 1.** UCLA LP-DOAS system setup. The left panel shows the side view of the field setup, with the DOAS instrument represented by the solid square and the retro-reflector arrays represented by the diamonds. Light paths are shown in solid lines (lower in black, middle in blue, and upper in red). Topography under each light path is shown in dashed lines of the same color. The right panel shows the top view of the light paths, with retro-reflectors located northwest of the DOAS instrument (white marker with black star).



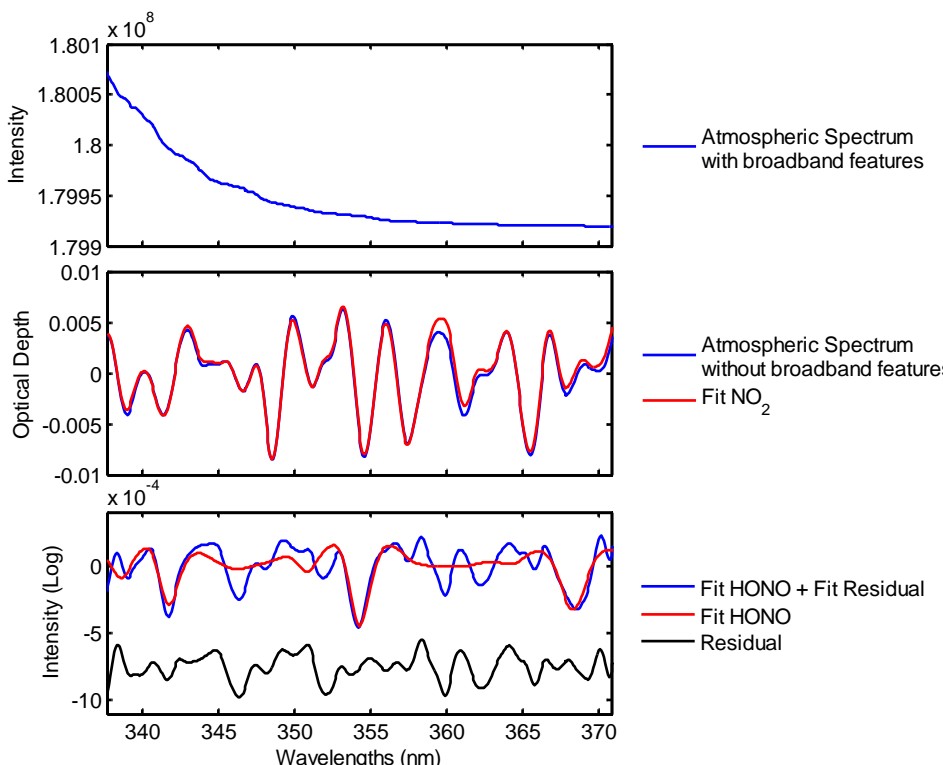

**Figure 2.** Result of a DOAS analysis of HONO and $NO_2$ of a spectrum measured on February 2, 2012 02:43 MST. a) Atmospheric spectrum with broadband features measured on the upper light path. b) $NO_2$ fit result (red line) superimposed on the atmospheric spectrum without broadband features (blue line). c) Fit result for HONO (red line) superimposed on sum of fitted HONO and unexplained residual structures in the fit (blue line); residual spectrum (black) shifted $7.5 \times 10^{-4}$ on the y-axis for better comparison with fitted HONO. The retrieved mixing ratio of HONO for this spectrum is: $144 \pm 22$ ppt.





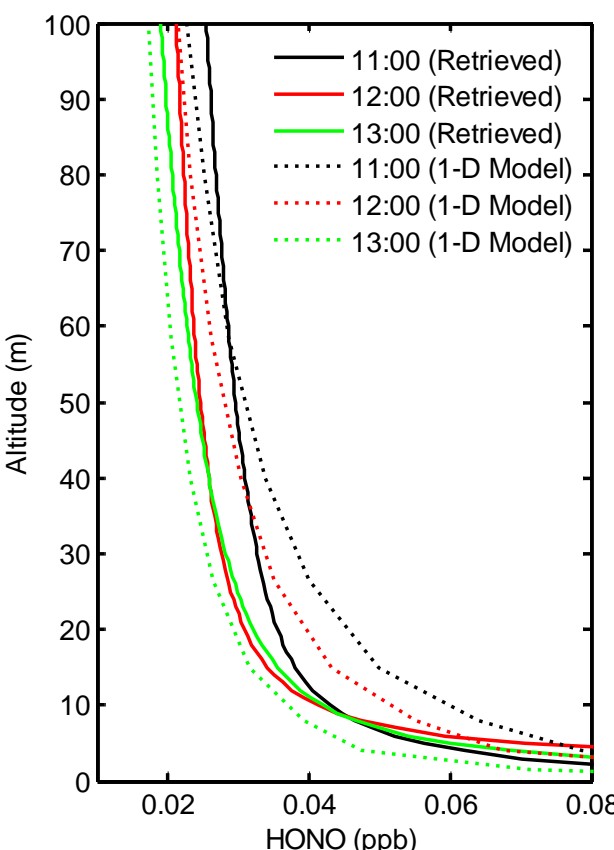

**Figure 3.** Hourly average of the vertical profiles of HONO retrieved using the least square fitting methodology (solid lines) for Feb 7, 2012 near noontime. The profile shape shows good agreement with those from vertical mixing ratio profiles of HONO modeled using the 1-D chemistry and transport model (dotted lines).





**Figure 4.** Overview of meteorological parameters (a,b) and mixing ratios of (c) NO$_2$, (d) O$_3$, and (e) HONO observed on the lower light path throughout the study.





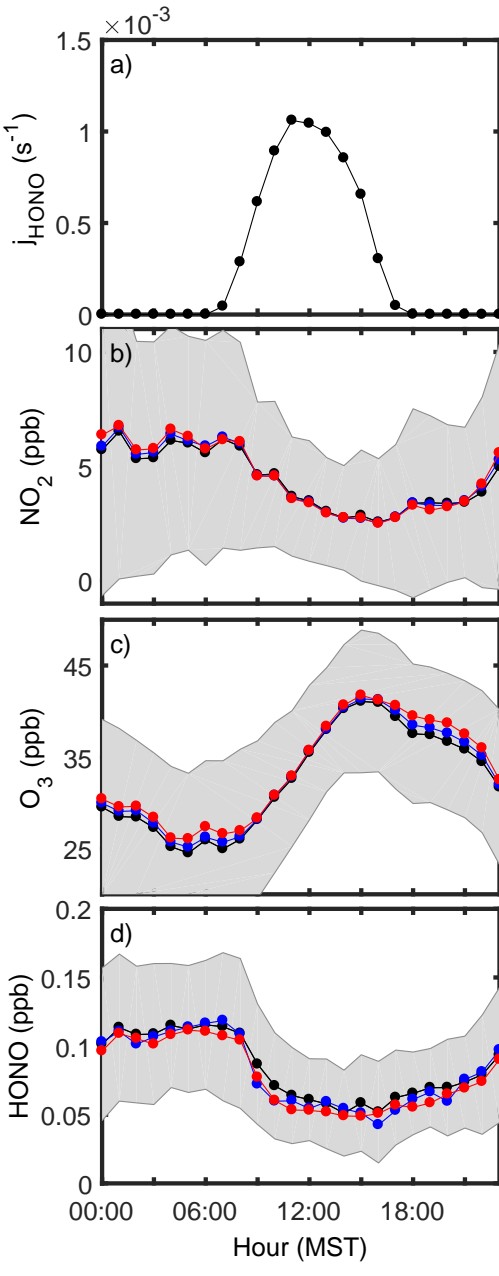

**Figure 5.** Sunny day hourly averages of the HONO photolysis frequency (a), and lower, middle, and upper LP-DOAS light path mixing ratios of (b) $NO_2$, (c) $O_3$, and (d) HONO during UBWOS 2012. The color coding of the light paths is given in Figure 1. The gray shading denotes the $1\sigma$ variability in the observations.





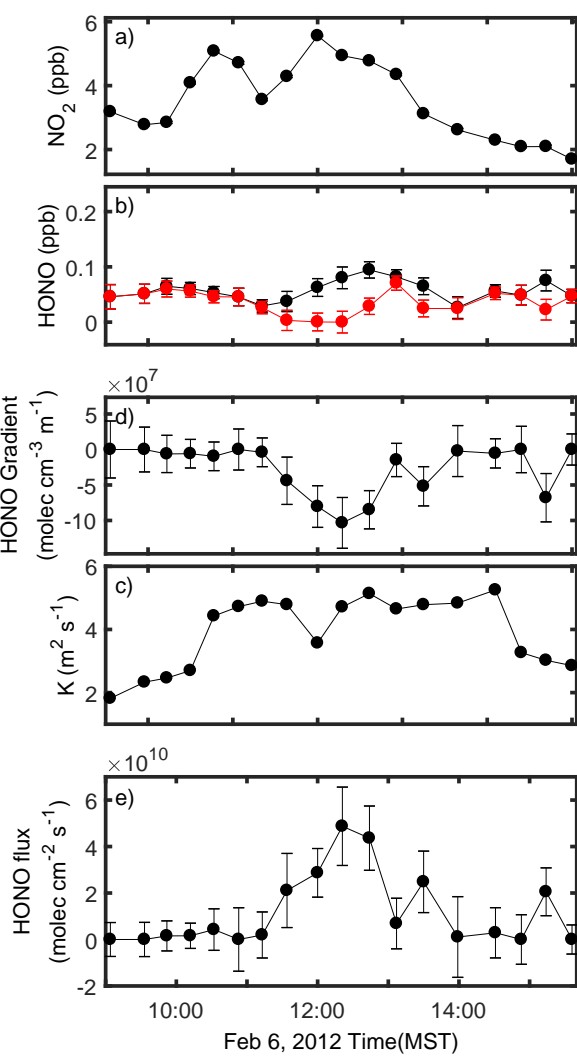

**Figure 6.** Mixing ratios of $NO_2$ on the lower light path (a) and HONO retrieved in the two height intervals (b). HONO concentration gradients calculated from the retrieved HONO mixing ratios (c) and observed eddy diffusivity coefficients (d) were used to calculate the vertical HONO flux on 6 February, 2012 (e).





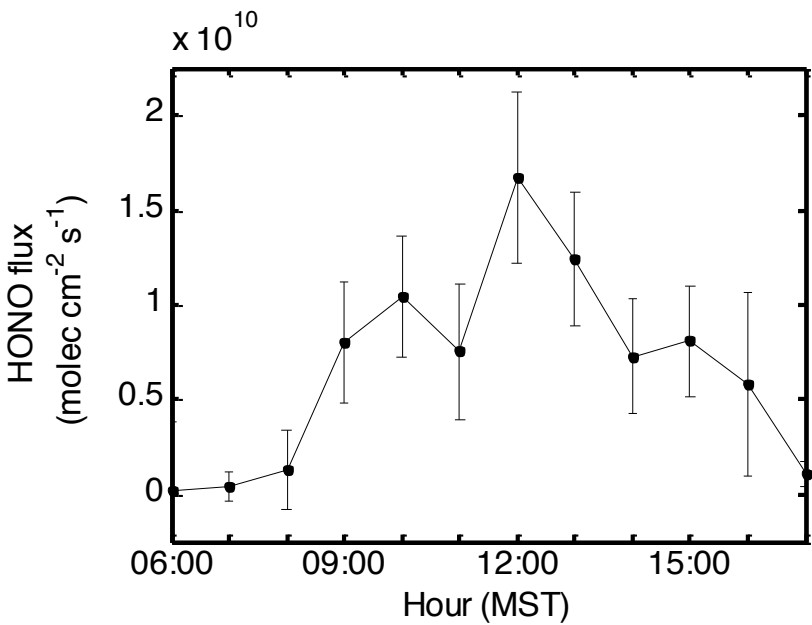

**Figure 7.** Sunny days hourly average vertical HONO fluxes derived from the HONO gradient observations during UBWOS 2012

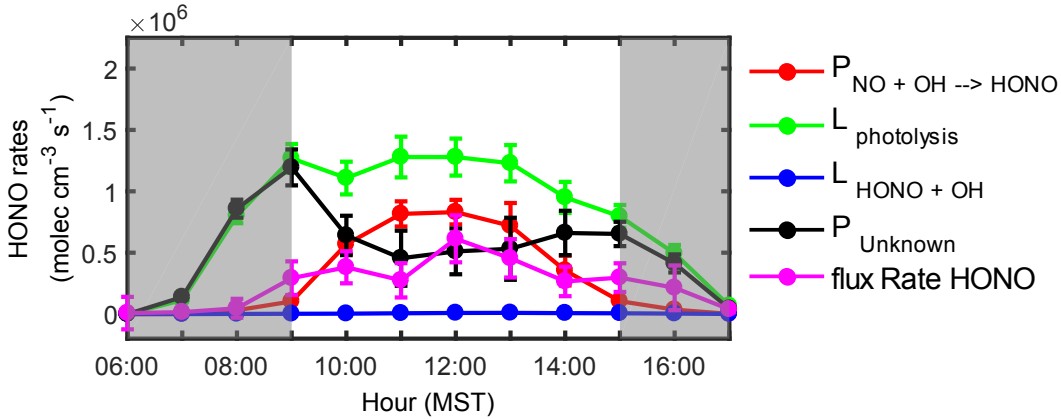

**Figure 8.** Sunny days hourly average HONO reaction rates. HONO production from OH + NO (red line, Equation 16), HONO photolysis (green line, Equation 17), HONO oxidation by OH (blue line, Equation 18), $P_{unknown}$ (black line, Equation 19) and HONO flux rate (magenta line). Grey boxes show the morning and evening time range when the pseudo-steady state assumption was not valid.



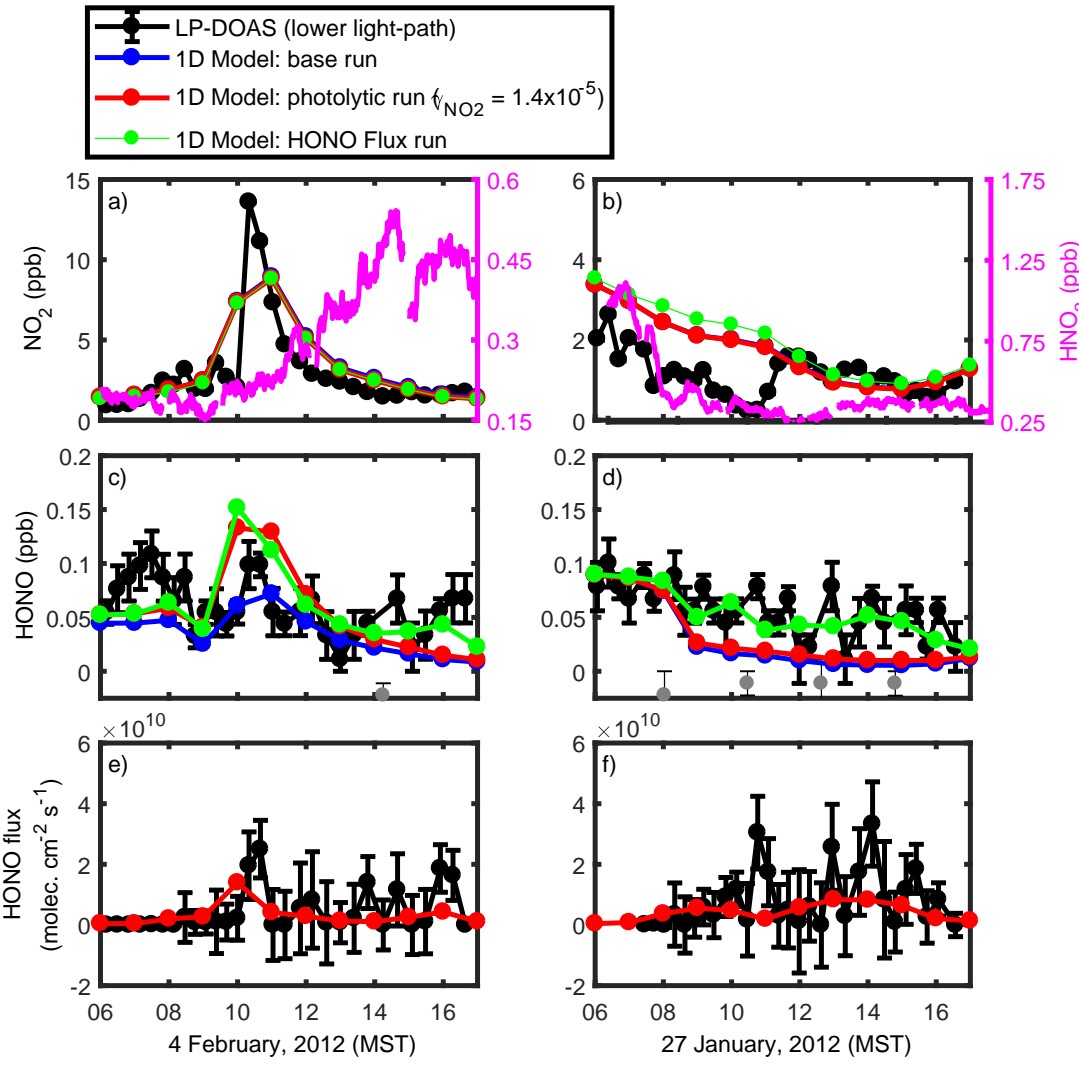

**Figure 9.** Comparison of modeled and measured $NO_2$ (a, b), HONO (c,d), and HONO flux (e,f) for 4 February 2012 (left panels: a, c, and e) and 27 January 2012 (right panels: b, d, and f). LP-DOAS lower light path $NO_2$ and HONO mixing ratios and errors are shown in black. We excluded a number of outliers (marked in grey) from the HONO analysis. These outliers are a consequence of problems with the DOAS analysis procedure for these specific measurement. Modeled $NO_2$ and HONO mixing ratios in the lower light path interval using the RCAT 8.2 base run are shown in blue, HONO flux run results are shown in green, and photolytic run results are shown in red (with $NO_2$ uptake coefficient of $1 \times 10^{-6}$ for 4 February and $1.4 \times 10^{-5}$ for 27 January). $HNO_3$ mixing ratios measured by NOAA are shown in magenta lines in the upper panels plots (a and b).





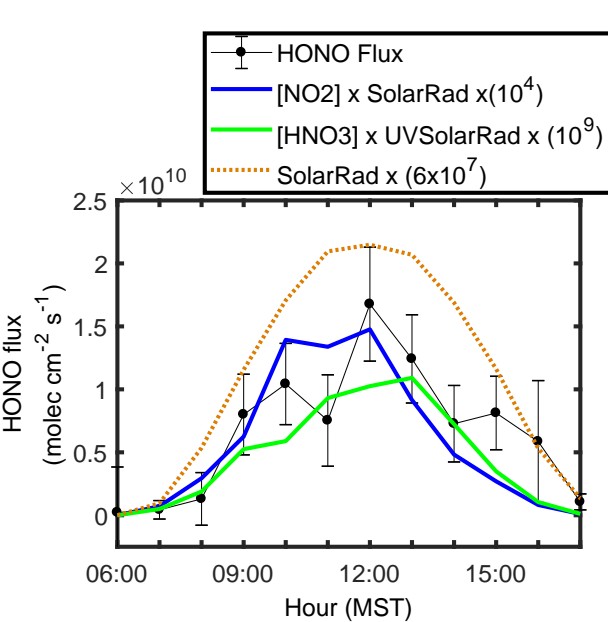

**Figure 10.** Sunny days hourly average HONO flux (black curve) is compared to solar radiation (dashed orange curve) and parametrization of the two most likely formation processes, photo-enhanced conversion of $NO_2$ (blue curve) and surface nitrate photolysis (green curve). The two parameterizations are scaled to match the general magnitude of the HONO flux.





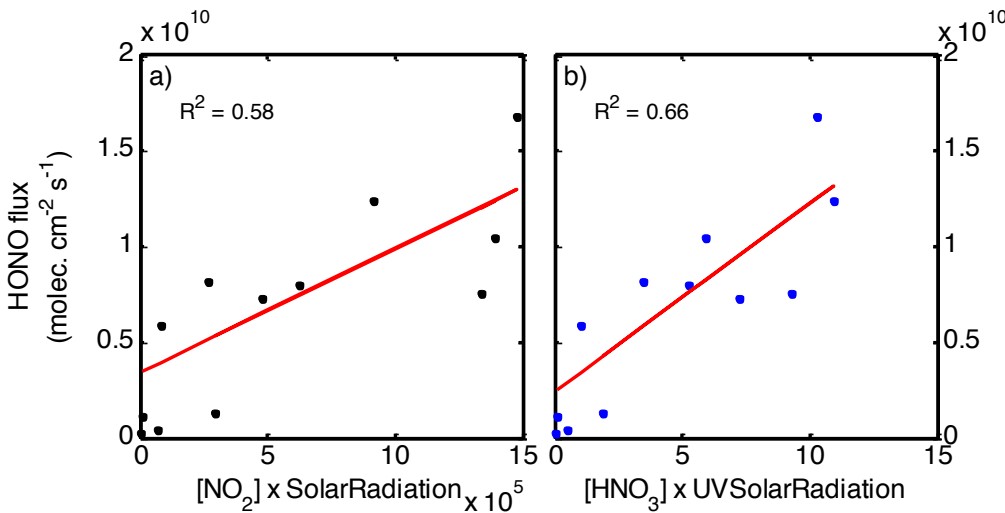

**Figure 11.** Correlation analysis of (a) HONO flux vs. [NO2] x solar radiation and (b) HONO flux vs. [HNO3] x UV solar radiation based on the hourly average sunny day diurnal profile.