# Peer review of "Nitrous acid formation in a snow-free wintertime polluted rural area"

_Atmospheric Chemistry and Physics, 2017_

## Referee Comment (RC1) · Anonymous Referee #1 · 3 Sep 2017

General Comments. The manuscript by Tsai et al. reports results of field measurements of nitrous acid (HONO) during a field campaign in the Uintah Basin during the winter of 2012. A long-path DOAS system measured atmospheric composition at three different altitude intervals, while a suite of accompanying measurement techniques provided trace gas and meterological data during the month-long field campaign. The data showed HONO gradients, with clear upward fluxes during the day. In addition, the authors were able to show that daytime HONO fluxes tracked solar irradiance. This rich dataset and powerful modeling analysis [1D chemical transport model (RCAT) and WRF-Chem] of the dataset provided opportunities to test out current hypotheses regarding HONO formation mechanisms. For example, one daytime event dominated by fresh emissions from a nearby coal-burning power plant where low HNO3 levels were

measured showed that photochemical NO2 to HONO conversion on soil surfaces was the dominant formation mechanism. Other occasions provided opportunities to test mechanisms involving surface nitrate photolysis, acid displacement of soil nitrite, and biological HONO releases.

The dataset acquired by the authors is unique and the study is quite novel. There are few studies published in the literature that focus on HONO formation during the wintertime. Moreover, the study was carried out in an arid region where snow was completely absent. This means the campaign provided a unique opportunity to study HONO chemistry over cold soil surfaces with little complication from vegetation surfaces. The study has some weakness and the authors are fully aware of the knowledge gaps and discuss them. For example, the study showed that daytime HONO production rates are correlated to both J(NO2) and J(HNO3) values, so there is still the question as to whether NO2 reduction by soil surfaces or nitrate photolysis (or both) are responsible for the surface daytime HONO source. However, the location and conditions of the field site, in addition to the quality and breadth of measurements made, and the use of models to interpret results and test hypotheses, enables insights into HONO formation mechanisms that were not previously possible in other studies. Most importantly, the study suggests that (at least for this region) surface photochemistry of NO2 and nitrate can account for the missing HONO source that is often cited in HONO studies. I know of few studies that have been able to show this so clearly.

The only major comment I have is regarding the likely speciation of nitrate as one of the HONO sources considered. The authors refer to HNO3(ads) or HNO3/nitrate(ads) photochemistry on soil surfaces as being responsible for a significant portion of the observed HONO, where (ads) refers to the adsorbed phase. While HNO3(g) is certainly present (it is measured by CIMS), it is highly unlikely that HNO3 is actually present as molecular nitric acid on soil surfaces in Uintah Basin. Nitric acid will dissociate upon contact with surfaces, even in the presence of minute amounts of water. In the presence of aluminosilicate soil minerals present in the region and at the alkaline soil pH

(NOAA measured a soil pH of 8.06) it is highly likely that hydrated nitrate (not molecular HNO3) is responsible for the observed HONO. The authors make the argument that HONO from microbial sources is not likely because of the alkaline soil and the fact that HONO's pKa is ∼3. Nitric acid's pKa is -1, so it is even less likely to be present in molecular form on surfaces. Realizing this may have some implications for some of the data analyses. For example, perhaps nitrate photochemistry parameters (absorption cross sections and quantum yields) may be more appropriate at describing the daytime HONO production rates than those of HNO3. It is clear that an estimate of the daytime surface flux of HONO due to nitrate is not possible without knowing nitrate densities at the soil surface. However, it may be worth noting that the correlation between HONO flux and [HNO3] x UV Solar Radiation shown in Figure 11 may be strong not because HNO3 is a direct photochemical source of HONO, but rather a source of surface nitrate (via deposition of HNO3) which is rather the direct source.

Minor Comments: [page, line]

[1, 13]: spell out 'agl' at first use.

[4, 15-17]: insert comma within numbers to denote place value – makes the numbers easier to read.

[6, 23]: replace "until" with "when"

[15, 15-21]: Consider including reference to Laufs et al. (ACP 2017, 17, 6907) who also showed recently that HONO production was correlated to [NO2]xJ(NO2).

[16, 1]: As per above comment, one may also have to consider [nitrate]xUV solar radiation.

[17, 12-14]: In discussing how much HONO is stored in soil could be released via strong acid displacement the next day, the authors state "...in this case one would expect that, as the surface reservoir is depleted throughout the night, the flux would also decrease..." Don't the authors mean during the day? It is the nitrite accumulated

during the nighttime that is the source of HONO during the day via the acid displacement mechanism. Presumably, displacement during the day would be what depletes the nitrite, while it accumulates during the nighttime.

[18, 6-8]: A sentence in quotes is given here. I assume it is quoted from Yan et al (2016) which appears earlier in the paragraph and somewhat removed from the sentence in question. I feel that the reference should again be referenced at the end of the paragraph to make that connection better.

[Figure 5D]: This figure is used to demonstrate that there is an elevation gradient of HONO concentration during the day, with higher concentrations at ground level compared to aloft. The mean values show this, although the differences are small due to the scale of the figure. In addition, the grey shading denoting 1-sigma variability are very large (spanning +/-0.05 ppb). Accounting for this variability, would not one conclude that there is no statistically significant difference between HONO concentration at the various altitudes? I realize the data are averages of all the sunny day data from the entire field campaign so there is tremendous variability. So, I can imagine the gradient is much more evident and variability insignificant if one restricts oneself to viewing data for a particular day. The authors may want to comment on this since at first glance with the plotted variability bands it may not be clear how one could say there are statistically significant differences in HONO for the various heights.

---

## Referee Comment (RC2) · Anonymous Referee #2 · 13 Sep 2017

Review on:

**Nitrous acid formation in a snow-free wintertime polluted rural area**
By Tsai et al.

**Summary:**

The authors present a very good study on missing photochemical HONO sources. They performed field measurements in the Uintah Basin, including HONO detection on 3 different heights. They observed higher concentrations near the ground and the determined flux followed the diel pattern of solar radiation (or photolysis frequency J). By performing budget analysis (pss calculation) a missing daytime HONO source was found. But the budget could be closed by adding the observed ground source. By correlation studies the authors conclude that a major part of HONO is formed on the ground either by light enhanced/induced heterogeneous reactions of NO2 or by photolysis of adsorbed HNO3 depending on the ambient NOx level. The study is underlined by model simulations (RCAT and WRF-Chem).

In general the manuscript is clearly written.

I suggest to publish the manuscript after the authors have addressed following minor comments:

**Concerning the understanding/scientific issues:**

Method part:
- NOx measurements: as you mentioned only the uncertainty/LOD of NO but later mainly discuss NO2 or NOx, please also add the detection limit of NO2. Specify the converter efficiency (what is the fraction of NO2 which is converted into NO during analysis, ~30% ??).
- P7 L 28: which CO instrument? You also note the techniques for the other compounds (GC/MS, Picarro, DOAS, …)
- I think the whole section is too long for the main manuscript. You could describe shortly the methods in the main manuscript and move the detailed description to a supplement.

Results/discussion:
- P12 L19-20 (fig 6b): I don´t understand why you are not using the measured mixing ratios here ("…calculated from retrieved HONO vertical profiles…"). Please explain!
- P14 L10: how long is the HONO lifetime, which time is too long to allow pss interpretation?
- P14 L23: Why not using the heterogeneous conversion rate (1.6% $h^{-1}$, e.g. Su et al., AE 42 (2008) 6219–6232) here, the 50% yield is only according to stoichiometry
- Fig 9: (e,f) are these the scaled fluxes?, please extent the capture;(b) why there is the discrepancy of the NO2 modeling on 27 Feb in the mornings (6-11 am) while the modelling in NO2 on Feb 4 is quiet good (even in the strong plume)?
- Fig 3: Can you explain why the model overpredicted the HONO at 11:00 and 12:00 and underestimated it at 13:00 – It is not a very good agreement (if only the shape is considered yes, but not the absolute values)

**Linguistic/graphic issues:**
- P2 L8: recycling of NOx, or NOx recycling
- P6 L15: cite correctly (Williams et al., year xy)
- P7 L16: the acronym RACM stands for Regional Atmospheric Chemistry Mechanism
- P7 L19: wrong bracket setting: … equation by Fuks and Sutugin (1971).

- P10 L23: …dominated by weak winds (delete periods)
- P15 L7: "leaf surface" said twice in one sentence – delete one time (… product of photolysis rate of HNO3 and nitrate loading on leaf surface… or add "adsorbed" HNO3…)

**References:**
- P17 L1-5. Wrong reference – this should be VandenBoer et al., 2015, not 2014?! In (2014) VandenBoer et al. suggested a ground reservoir but don´t argued with acid displacement.
- In the list: Wong et al., 2011 not correct cited – Journal name missing, check also other references if form is consistent

**Figures:**
- Fig. 4: Please draw clearer – especially in the HONO plot, dots are hardly to distinguish (maybe use lines instead, or remove error bars?) – what do the error bars mean?
- Sometimes you use left or right panel in the capture but in the figure you label it with a and b – so please also refer to a,b in the capture (fig 2: label single plots as a,b,c – as written in the capture)
- Fig 5/6: why not also explain the colors here?
- Fig 6/7/8/9: meaning of error bars
- Fig 9 (b) the HNO3 layer is shifted
- MST = local time????, please explain

---

## Author Comment (AC1) · 17 Nov 2017

**Response to Reviewers Comments on manuscript acp-2017-648 entitled: "Nitrous acid formation in a snow-free wintertime polluted rural area", by Tsai et al.**

We would like to thank both reviewers for their helpful comments. Below are our responses (in italic) to the comments.

**Response to Reviewer 1:**

The only major comment I have is regarding the likely speciation of nitrate as one of the HONO sources considered. The authors refer to HNO3(ads) or HNO3/nitrate(ads) photochemistry on soil surfaces as being responsible for a significant portion of the observed HONO, where (ads) refers to the adsorbed phase. While HNO3(g) is certainly present (it is measured by CIMS), it is highly unlikely that HNO3 is actually present as molecular nitric acid on soil surfaces in Uintah Basin. Nitric acid will dissociate upon contact with surfaces, even in the presence of minute amounts of water. In the presence of aluminosilicate soil minerals present in the region and at the alkaline soil pH (NOAA measured a soil pH of 8.06) it is highly likely that hydrated nitrate (not molecular HNO3) is responsible for the observed HONO.

The authors make the argument that HONO from microbial sources is not likely because of the alkaline soil and the fact that HONO's pKa is _3. Nitric acid's pKa is -1, so it is even less likely to be present in molecular form on surfaces. Realizing this may have some implications for some of the data analyses. For example, perhaps nitrate photochemistry parameters (absorption cross sections and quantum yields) may be more appropriate at describing the daytime HONO production rates than those of HNO3. It is clear that an estimate of the daytime surface flux of HONO due to nitrate is not possible without knowing nitrate densities at the soil surface. However, it may be worth noting that the correlation between HONO flux and [HNO3] x UV Solar Radiation shown in Figure 11 may be strong not because HNO3 is a direct photochemical source of HONO, but rather a source of surface nitrate (via deposition of HNO3) which is rather the direct source.

*The nature of surface adsorbed HNO$_3$/nitrate and the chemical mechanism that leads to HONO formation upon photolysis is an open scientific question (see for example a recent manuscript by Ye et al, ES&T 2016 that studies surface adsorbed HNO$_3$/nitrate photolysis) that we cannot adequately address based on our field data. We completely agree with the reviewer that from a bulk soil perspective, with a basic pH, one would expect that all deposited HNO$_3$ is in the form of nitrate and that surface deposition of HNO$_3$ is likely an important source of surface HNO$_3$/nitrate.*

*However, the ground surface is complex and does not necessarily behave like the bulk soil. For example, insoluble surfaces (rocks, plants, etc,) will not behave like the bulk soil. The low temperatures, and often dry conditions, will also lead to a quasi liquid layer on soil grains that may not behave like a bulk solution. Finally, only the top of the soil is well enough illuminated to allow HNO$_3$/nitrate photolysis and the bulk soil pH measurement is not necessarily representative for this uppermost layer. This layer is where most of the HNO$_3$ (and other atmospheric acids) will be deposited, so one could expect that HNO$_3$/nitrate levels are higher and thus pH lower.*

*With all these uncertainties, we do not feel that it is appropriate to classify HNO$_3$/nitrate as surface nitrate in our study. We have therefore left the naming unchanged in the manuscript, to reflect our lack of knowledge on the nature of surface adsorbed HNO$_3$/nitrate.*

*The correlation between UV solar radiation x [HNO$_3$] and HONO flux in Figure 11 is based on the assumption that measured gas-phase HNO$_3$ is a proxy of the surface HNO$_3$/nitrate that leads to HONO*

*formation (see page 15 line 27-28). We do not propose in the paper that gas-phase HNO$_3$ photolysis is an important source, i.e. Page 16 line 30 clearly states: "The strong correlation between HONO flux and [HNO$_3$] x UV solar radiation suggest that photolysis of surface HNO$_3$/nitrate might explain the observed HONO flux."*

Minor Comments: [page, line]

[1, 13]: spell out 'agl' at first use.

*done*

[4, 15-17]: insert comma within numbers to denote place value – makes the numbers easier to read.

*Changed number to " 50 thousand " and " 10 thousand " according to AMT formatting instructions. Please note that commas are used in some countries as the decimal point and adding a comma as a place value could thus be confusing.*

[6, 23]: replace "until" with "when"

*done*

[15, 15-21]: Consider including reference to Laufs et al. (ACP 2017, 17, 6907) who also showed recently that HONO production was correlated to [NO2]xJ(NO2).

*The following sentence was added to the manuscript to cite this paper: "A strong correlation of the upwards HONO flux over agricultural fields with the product of NO$_2$ concentration with the NO$_2$ photolysis rate was recently also reported by Laufs et al. (2017)."*

[16, 1]: As per above comment, one may also have to consider [nitrate]xUV solar radiation.

*This would have been a better choice, but since we did not directly observe nitrate concentrations, but are rather using observed [HNO3] as a proxy for nitrate, it seems clearer to refer to the observed nitric acid concentration in this sentence.*

[17, 12-14]: In discussing how much HONO is stored in soil could be released via strong acid displacement the next day, the authors state ". . .in this case one would expect that, as the surface reservoir is depleted throughout the night, the flux would also decrease. . ." Don't the authors mean during the day? It is the nitrite accumulated during the nighttime that is the source of HONO during the day via the acid displacement mechanism. Presumably, displacement during the day would be what depletes the nitrite, while it accumulates during the nighttime.

*Thanks for pointing out this mistake. We have changed "night" to "day" in this sentence.*

[18, 6-8]: A sentence in quotes is given here. I assume it is quoted from Yan et al (2016) which appears earlier in the paragraph and somewhat removed from the sentence in question. I feel that the reference should again be referenced at the end of the paragraph to make that connection better.

*The reference was added to this sentence to clearly cite the paper.*

[Figure 5D]: This figure is used to demonstrate that there is an elevation gradient of HONO concentration during the day, with higher concentrations at ground level compared to aloft. The mean values show this, although the differences are small due to the scale of the figure. In addition, the grey shading denoting 1-sigma variability are very large (spanning +/-0.05 ppb). Accounting for this variability, would not one conclude that there is no statistically significant difference between HONO concentration at the various altitudes? I realize the data are averages of all the sunny day data from the entire field campaign so there is tremendous variability. So, I can imagine the gradient is much more evident and variability insignificant if one restricts oneself to viewing data for a particular day. The authors may want to comment on this since at first glance with the plotted variability bands it may not be clear how one could say there are statistically significant differences in HONO for the various heights.

*Figure 5 shows the sunny day light-path averaged mixing ratios, i.e. the data before the concentration profiles retrieval, for each of the three light paths, not the concentration profiles. Following a comment by Reviewer 2 we have included language in Section 2.4 that better explains this quantity: "As illustrated in Figure 1, the LP-DOAS observes trace gas concentrations averaged over the different light paths, i.e. averaged from the ground to the reflector height of each light path."*

*The gray shading does not represent the statistical error of the observations, but rather the spread of the data expressed as the $1\sigma$ standard deviation. We have revised the last sentence in the caption of Figure 5 to explain this: "The gray shading denotes the spread of the observed light-path averaged mixing ratios expressed as a $1\sigma$ variability of the hourly data."*

*The reviewer is correct that the HONO gradients are much more pronounced, and statistically relevant, on single days in the retrieved concentration profiles, as for example shown in Figure 6. The purpose of Figure 5 is, however, to show the averaged diurnal hourly mixing ratios of all trace gases, that the gradients are already visible in the raw light-path averaged data, and that the behavior of the gradients of all three gases are consistent with the expectations.*

**Response to Reviewer 2:**

Concerning the understanding/scientific issues:

Method part:
- NOx measurements: as you mentioned only the uncertainty/LOD of NO but later mainly discuss NO2 or NOx, please also add the detection limit of NO2. Specify the converter efficiency (what is the fraction of NO2 which is converted into NO during analysis, ~30% ??).

*Only the NO observations from the in-situ NOx analyzer are used in the manuscript. Reported $NO_2$ mixing ratios are always from the LP-DOAS observations. To keep this section short we therefore did not discuss the $NO_2$ detection limits from this instrument. We deleted the sentence explaining the photolytic converter, as it was not relevant for our study, and added the following sentence to clarify this: "Only the NO observations from the $NO_x$ instruments were used in this study".*

- P7 L 28: which CO instrument? You also note the techniques for the other compounds (GC/MS, Picarro, DOAS, …)

*We added the following sentence to Section 2.2.4 and renamed this section "CO and VOC measurements": "Carbon monoxide (CO) was measured with a commercial vacuum–UV resonance fluorescence instrument; the accuracy and precision of the CO measurements were ±4\% and ±0.5 ppbv respectively."*

- I think the whole section is too long for the main manuscript. You could describe shortly the methods in the main manuscript and move the detailed description to a supplement.

*The 1D chemistry and transport model is crucial to support the choice of the HONO concentration profile shape for the retrieval, as well as the interpretation of the data in Section 3.6. In addition, our 1D model is different from other models as it is based on the simulation of molecular processes, i.e. reactive conversion of $NO_2$ at the surface. This section describes both the model as well as the initialization approach used in the remainder of the study. We believe that the description is already very concise and moving it to the supplement will weaken the line of arguments presented in the manuscript. We have therefore kept this section in its original form.*

Results/discussion:

- P12 L19-20 (fig 6b): I don´t understand why you are not using the measured mixing ratios here ("…calculated from retrieved HONO vertical profiles…"). Please explain!

*It is not possible to directly use the observations from the LP-DOAS because the instrument measures concentrations averaged from the ground to the height of the reflector. In addition, we face a situation where the ground is slightly inclined between the LP-DOAS telescope and the reflectors (see Figure 1). As this did not seem to be as clear as we had hoped, we revised the first paragraph in Section 2.4:*

*"As illustrated in Figure 1, the LP-DOAS observes trace gas concentrations averaged over the different light paths, i.e. averaged from the ground to the reflector height of each light path. It is thus necessary to convert these vertically averaged trace gas observations to vertical concentrations gradients/profiles. This is typically achieved using a geometric approach, where the ground is considered flat (Stutz et al., 2014, Wang et al. 2006, Wong et al., 2012,2013, Tsai et al., 2014)."*

- P14 L10: how long is the HONO lifetime, which time is too long to allow pss interpretation?

*The typical HONO lifetime was listed on page 11 line 24: " … the short photolytic lifetime of HONO (about 13 minutes near noontime and less than 21 minutes between 09:00 and 15:00 LT)". The question of what length is too long for a pseudo-steady-state interpretation is difficult to answer. If one solely considers the instantaneous chemical formation of HONO, i.e. starting at [HONO]=0, one would expect that it will take 3-5 times of the photolytic lifetime of HONO to be close to the PSS. However, the situation is different when we are considering a slowly varying system, where the source and sink rates of HONO are slowly changing throughout the day, i.e. the system is nudged a little bit away from the PSS at every time step. In this case the PSS is never reached but the chemical system is always close to the PSS. A PSS calculation can therefore only be used as a best estimate. This is one of the reasons why we included the 1D chemical transport model calculation, which are not based on the PSS approach. As shown in Figure 9 and discussed in Section 3.6, driving the 1D model with the observed fluxes leads to HONO concentrations close to the observations.*

- P14 L23: Why not using the heterogeneous conversion rate (1.6% h-1, e.g. Su et al., AE 42 (2008) 6219–6232) here, the 50% yield is only according to stoichiometry

*Our 1D model directly uses the molecular uptake of $NO_2$ on the surface with an uptake coefficient of $2x10^{-5}$ to determine the dark formation rate of HONO. In addition, we also consider the loss of HONO on the surface with a reactive uptake coefficient of $10^{-3}$ (see Section 2.3.1). The vertical transport of the HONO formed or lost, i.e. the balance of these two effects, at the surface is then explicitly calculated to determine the HONO vertical profile. We have shown the success of this approach in Wong et al 2011, 2013, where we modeled nocturnal and daytime HONO concentration profiles. The approach of Su et al, 2008 is a much more parameterized description of this process, which would not allow us to model vertical concentration profiles. In addition, our model has the advantage that it clearly separates the chemical processes, albeit also parameterized using uptake coefficients and stoichiometry, from the vertical transport processes and molecular collisions with the surface.*

- Fig 9: (e,f) are these the scaled fluxes?, please extent the capture;(b) why there is the discrepancy of the NO2 modeling on 27 Feb in the mornings (6-11 am) while the modelling in NO2 on Feb 4 is quiet good (even in the strong plume)?

*These are the fluxes at 19m altitude that result from the scaled fluxes. Thanks for pointing this out. It was not clearly explained in the text. We added a sentence in Section 2.4.2 stating: "HONO gradients and fluxes were calculated at 19 m agl." We also changed "HONO flux" to "HONO flux at 19m agl in the caption of Figure 9. In addition, we added a sentence in Section 3.6 further clarifying what is shown in Figure 9 e,f: "Figure 9 e, f show that the modeled HONO flux at 19 m agrees well with the observations after this scaling step. '*
*We added the following explanation to point out the limitations of our 1D modeling approach in simulating $NO_2$ in situations with changing air masses: "The simulations (Figure 9 a,b) generally reproduce $NO_2$ mixing ratios well, except during advection events. For example, on February 4 between 10:00 and 12:00 the model has difficulties reproducing the fast concentration changes associated with the passing of a power plant plume. Similarly, the drop in $NO_2$ in the morning of January 27 is likely due to a change in air mass, which is not captured well by our 1D model. "*

- Fig 3: Can you explain why the model overpredicted the HONO at 11:00 and 12:00 and underestimated it at 13:00 – It is not a very good agreement (if only the shape is considered yes, but not the absolute values)

*Modeled concentrations at 11:00 and 12:00 are larger at lower altitudes and smaller at higher altitudes (including above 100m). This behavior is expected if the vertical transport in the model is somewhat smaller than in the atmosphere. This was explained in Section 2.4.1 (Page 9 line 16): "The slight differences between the retrieved and model profiles at 11:00 and 12:00 LT can be explained by the uncertainty in the vertical mixing parameters used in the model, i.e. stronger vertical mixing in the model leads to modeled profiles which are less steep than the one retrieved from the observations.".*
*It should also be pointed out that the purpose of this figure is to convince the reader that the parameterization of the profile shape used in the retrievals is reasonable, i.e. the focus here is really on the shape rather than the magnitude.*

Linguistic/graphic issues:

- P2 L8: recycling of NOx, or NOx recycling

*corrected*

- P6 L15: cite correctly (Williams et al., year xy)

*corrected*

- P7 L16: the acronym RACM stands for Regional Atmospheric Chemistry Mechanism

*Thanks for finding this mistake. We have added the word "Chemistry"*

- P7 L19: wrong bracket setting: … equation by Fuks and Sutugin (1971).

*corrected.*

- P10 L23: …dominated by weak winds (delete periods)

*done*

- P15 L7: "leaf surface" said twice in one sentence – delete one time (… product of photolysis rate of HNO3 and nitrate loading on leaf surface… or add "adsorbed" HNO3…)

*corrected*

**References:**
- P17 L1-5. Wrong reference – this should be VandenBoer et al., 2015, not 2014?! In (2014) VandenBoer et al. suggested a ground reservoir but don´t argued with acid displacement.

*We corrected this mistake.*

- In the list: Wong et al., 2011 not correct cited – Journal name missing, check also other references if form is consistent

*done*

Figures:

- Fig. 4: Please draw clearer – especially in the HONO plot, dots are hardly to distinguish (maybe use lines instead, or remove error bars?) – what do the error bars mean?

*We made the figure clearer and added the following sentence to the caption: "Error bars in the trace gas panels (a-e) denote the statistical error of each observation derived by the DOAS analysis. Please note that the $NO_2$, and $O_3$ error bars are smaller than the data markers."*

- Sometimes you use left or right panel in the capture but in the figure you label it with a and b – so please also refer to a,b in the capture (fig 2: label single plots as a,b,c – as written in the capture)

- Fig 5/6: why not also explain the colors here?

*We added the following language to the caption of figure 5 : "i.e. lower light path in black, middle light path in blue, and upper light path in red", and "(black: lower interval, red: upper interval)" to the caption of Figure 6.*

- Fig 6/7/8/9: meaning of error bars

*We added "Error bars show the $1\sigma$ uncertainty of the respective data points." to each of the captions.*

- Fig 9 (b) the HNO3 layer is shifted

*corrected*

- MST = local time????, please explain

*Thanks for pointing this out. The manuscript was inconsistent in how it named times. We have corrected this. Now all times in the text are also in Mountain Standard Time (MST). At the first mention of MST in the text we added the following explanation (page 9, line 18): "…MST (Mountain Standard Time, which is the local time zone in Utah and will be used throughout this paper)…"*
*We also re-introduced the abbreviation in the caption of Figure 2: "Mountain Standard Time (MST)"*